# ARTICULATE-ANYTHING:
# AUTOMATIC MODELING OF ARTICULATED OBJECTS VIA A VISION-LANGUAGE FOUNDATION MODEL

**Long Le, Jason Xie, William Liang, Hung-Ju Wang, Yue Yang, Yecheng Jason Ma,
Kyle Vedder, Arjun Krishna, Dinesh Jayaraman, Eric Eaton**
University of Pennsylvania

https://articulate-anything.github.io/

## ABSTRACT

Interactive 3D simulated objects are crucial in AR/VR, animations, and robotics, driving immersive experiences and advanced automation. However, creating these articulated objects requires extensive human effort and expertise, limiting their broader applications. To overcome this challenge, we present ARTICULATE-ANYTHING, a system that automates the articulation of diverse, complex objects from many input modalities, including text, images, and videos. ARTICULATE-ANYTHING leverages vision-language models (VLMs) to generate code that can be compiled into an interactable digital twin for use in standard 3D simulators. Our system exploits existing 3D asset datasets via a mesh retrieval mechanism, along with an actor-critic system that iteratively proposes, evaluates, and refines solutions for articulating the objects, self-correcting errors to achieve a robust outcome. Qualitative evaluations demonstrate ARTICULATE-ANYTHING's capability to articulate complex and even ambiguous object affordances by leveraging rich grounded inputs. In extensive quantitative experiments on the standard PartNet-Mobility dataset, ARTICULATE-ANYTHING substantially outperforms prior work, increasing the success rate from 8.7–12.2% to 75% and setting a new bar for state-of-the-art performance. We further showcase the utility of our system by generating 3D assets from in-the-wild video inputs, which are then used to train robotic policies for fine-grained manipulation tasks in simulation that go beyond basic pick and place. These policies are then transferred to a real robotic system.

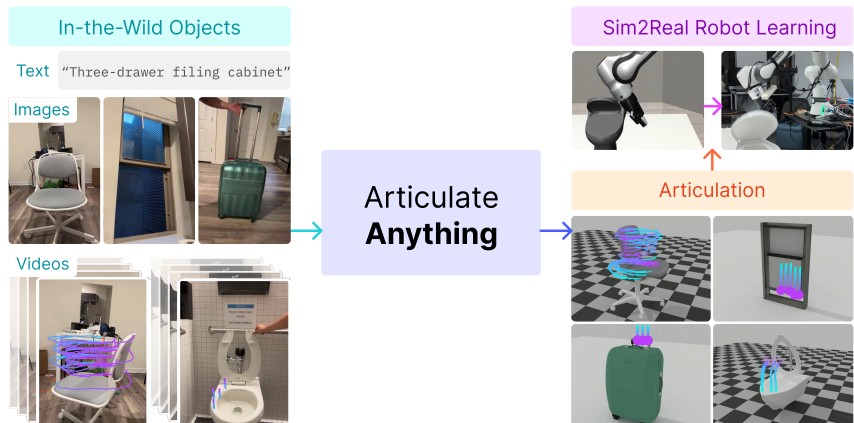

Figure 1: Given text, images, or videos showing an object's motion, ARTICULATE-ANYTHING automatically generates its 3D interactable digital twin, handling a wide variety of objects and affordances. Among other applications, these articulated assets can be used to train robotic manipulation policies in Sim2Real. Full video demonstrations and source code are available on the website.

## 1 INTRODUCTION

One of the most promising avenues in the quest for generally capable robots is the pursuit of scaling robot training in simulation for deployment in the real world. We have developed fast, accurate

simulators that scale to millions of FPS and hundreds of GPUs (Xiang et al., 2020; Makoviychuk et al., 2021), enabling policy learning on a staggering scale. However, a critical bottleneck in this research direction persists: the immense human labor required to construct realistic, *interactable* environments for these agents to learn within. Despite the existence of large, open libraries of static object geometries — with the largest open dataset containing over 10 million objects (Deitke et al., 2024) — we have comparatively minuscule open libraries of *articulated* 3D objects (only around 2,300 objects (Xiang et al., 2020)). This scarcity stems from the time-consuming, labor-intensive, and expertise-demanding nature of the manual annotation process.

To address this challenge, we present ARTICULATE-ANYTHING, a novel approach in automatic articulation that harnesses the power of leading foundation vision-language models (VLMs) to articulate a diverse range of objects of arbitrary complexity through iterative feedback (Fig. 1). ARTICULATE-ANYTHING represents a step function improvement in quality, accuracy (8.7-12.2% to 75%), and generalizability over prior art (Chen et al., 2024; Mandi et al., 2024), overcoming previous limitations that restricted success to only a narrow range of object categories and joint types. Unlike prior art, which has been limited by the impoverished input of bounding boxes or static images, ARTICULATE-ANYTHING affords the flexibility of consuming rich, grounded inputs from text, images, or even videos, enabling users to request exotic articulation descriptions or resolve articulation ambiguities. For example, the right column of Fig. 7 features a digital model of a window that could plausibly slide or tip to open; when ARTICULATE-ANYTHING is shown an in-the-wild video demonstration, it accurately produces the desired sliding motion.

To achieve this level of flexibility and accuracy, ARTICULATE-ANYTHING employs an actor-critic system with two core components: (1) a vision-language actor that synthesizes high-level Python code, which can be compiled into Unified Robot Description Format (URDF) files and (2) a vision-language critic that provides feedback on the rendered prediction compared against available ground-truth. The result is an agentic system that can automatically self-evaluate and iteratively improve the articulation of complex objects. Beyond robotics, the flexibility of ARTICULATE-ANYTHING's inputs married with its high-quality outputs puts automatic generation of rich, high-quality, and diverse virtual environments within reach with broad-reaching applications to 3D/VR (Kim et al., 2024), human-computer interaction (Jiang et al., 2023), and animation (Yang et al., 2022).

Our key contributions include:

1. **ARTICULATE-ANYTHING**: We present a vision-language actor-critic system that accurately articulates objects from diverse input modalities, including texts, images, and videos.
2. **Articulation as program synthesis:** We develop a high-level Python API compilable into URDFs, enabling the VLM actor to generate compact, easily debuggable programs.
3. **Visually grounded inputs enable closed-loop iterative refinement:** We highlight the critical role of grounded visual inputs in articulating ambiguous and complex objects, leveraging such inputs in a closed-loop system that self-evaluates and self-improves its articulations.
4. **Extensive evaluation demonstrates superior performance:** Prior works in automatic articulation are only evaluated on a handful of object categories. Quantitative analysis on the entire PartNet-Mobility dataset shows dramatic improvement over existing methods, increasing the success rate from 8.7–12.2% of prior work to 75%.

## 2 RELATED WORK

**3D Asset Datasets.** Large open datasets of 3D objects, such as 3D Warehouse and Objaverse (Deitke et al., 2024), provide extensive collections of 3D models uploaded by independent artists. However, most objects in these repositories lack articulation. Datasets like ShapeNet (Chang et al., 2015) and PartNet (Mo et al., 2018) decompose objects into parts (links) but do not include kinematic joints. Some datasets, such as PartNet-Mobility (Xiang et al., 2020) and GAPartNet (Geng et al., 2022), do feature articulated objects with links and joints, but these annotations are manually created so their number of objects is modest. Our work aims to automate the creation of articulated 3D assets suitable for robotic tasks, reducing the reliance on manual annotation.

**Articulated Object Modeling.** Articulated object modeling is a significant area of research, encompassing tasks such as perception, reconstruction, and generation (Liu et al., 2024). In perception, works like (Zeng et al., 2021; Hu et al., 2017) focus on identifying and understanding articulated objects. Reconstruction efforts from single RGB images (Chen et al., 2024), RGBD images (Weng

et al., 2024), multi-view images (Liu et al., 2023a), or point clouds (Jiang et al., 2022) aim to rebuild articulated models. Generation methods, including those leveraging connectivity graphs (Liu et al., 2023b) or neural approaches (Lei et al., 2023), focus on creating new articulated objects. Our goal is to develop an end-to-end pipeline that spans from perception to reconstruction and generation, utilizing intuitive and easily accessible inputs from humans, such as language or videos, instead of relying on more specialized modalities, such as point clouds or graphs.

## 3 PROBLEM FORMULATION

To ensure compatibility with standard 3D simulators, we represent 3D models in the Unified Robot Description Format (URDF). In URDF, an object is structured in a hierarchical tree consisting of nodes (links) and edges (joints). The links represent parts of an object (e.g., drawers and doors of a kitchen island), and joints specify how each part moves. The two most common joints are prismatic, representing a translational movement (e.g., a sliding drawer), and revolute, representing a rotation (e.g., a pivoting door).

Given an input $x$ of text, image, or video depicting an object, we assume there exists a ground-truth URDF $u^*$ (e.g., one that a human could create). The goal of articulation is to automatically construct a URDF model $\hat{u}$ that comprises the same set of links as $u^*$ while minimizing the difference between URDF models via the following loss:

$$\mathcal{L}_{\text{total}} = \mathcal{L}_{\text{mesh}} + \mathcal{L}_{\text{link}} + \mathcal{L}_{\text{joint}} \;, \tag{1}$$

where $\mathcal{L}_{\text{mesh}} = \sum_{i=1}^{N} \text{Chamfer}(\hat{P}_i, P_i^*)$ measures the discrepancy in visual appearances between the predicted and ground-truth links using point clouds $\hat{P}_i$ and $P_i^*$. $N$ is the number of links. $\mathcal{L}_{\text{link}}$ captures 3D pose differences between predicted and ground-truth links:

$$\mathcal{L}_{\text{link}} = \sum_{i=1}^{N} \underbrace{\|\hat{\mathbf{x}}_i - \mathbf{x}^*{}_i\|_2}_{\text{position error}} + \underbrace{2 \arccos(|\hat{q}_i \cdot q_i^*|)}_{\text{orientation error}} \;, \tag{2}$$

where $\mathbf{x}_i$ is a 3D coordinate and $q_i$ is a quaternion. $\mathcal{L}_{\text{joint}}$ quantifies the joint discrepancies.

$$\mathcal{L}_{\text{joint}} = \mathcal{L}_{\text{joint type}} + \mathcal{L}_{\text{joint axis}} + \mathcal{L}_{\text{joint origin}} + \mathcal{L}_{\text{joint limit}} \;, \tag{3}$$

where $\mathcal{L}_{\text{joint type}}$ is a cross entropy loss, $\mathcal{L}_{\text{joint axis}}$ is the angular difference between axes, $\mathcal{L}_{\text{joint origin}}$ is the 3D pose difference between joint origins, and $\mathcal{L}_{\text{joint limit}}$ measures the difference in range and direction of motion. In practice, jointly optimizing this objective becomes intractable. Consequently, we instead tackle each loss sequentially, decomposing the problem into three phases: (1) Mesh Retrieval, (2) Link Placement, and (3) Joint Prediction. Furthermore, access to $u^*$ is often limited, making traditional optimization methods for computing these losses impractical. To overcome this, we leverage vision-language model (VLM) agents to approximate the loss functions directly from visual inputs. Additionally, instead of refining solutions through a conventional optimizer, VLM agents are also used to propose solutions in the form of code and iteratively refine them based on feedback from a critic module. This process is detailed in the following sections.

## 4 ARTICULATED OBJECT GENERATION VIA ACTOR-CRITIC VLMS

In this section, we introduce ARTICULATE-ANYTHING, a system capable of generating articulated objects from various input modalities. Figure 2 presents an overview of our method, which includes three main components: (1) Mesh Retrieval (Sec. 4.1), which reconstructs the 3D structure for each object part by retrieving meshes from a 3D asset library, (2) Link Placement (Sec. 4.2), which spatially arranges the parts, and (3) Joint Prediction (Sec. 4.3), which determines the potential kinematic movements between parts. Additionally, we describe an optional step, targeted affordance extraction, in Sec. 4.4. Our system consists of many specialized VLM agents. Each agent is a Google's Gemini Flash-1.5 visual language model (Team et al., 2023) prompted with no more than 20 in-context examples to perform different tasks. All system and task instruction prompts are provided in the source code.

Crucially, our system produces high-level Python code compilable into URDFs via a custom API we have developed. Directly generating URDFs is challenging due to (1) the verbosity of URDF/XML, which increases the likelihood of hallucination in long contexts, and more importantly, (2) the need for performing complex mathematics in-place in many cases. For instance, articulating a rotation

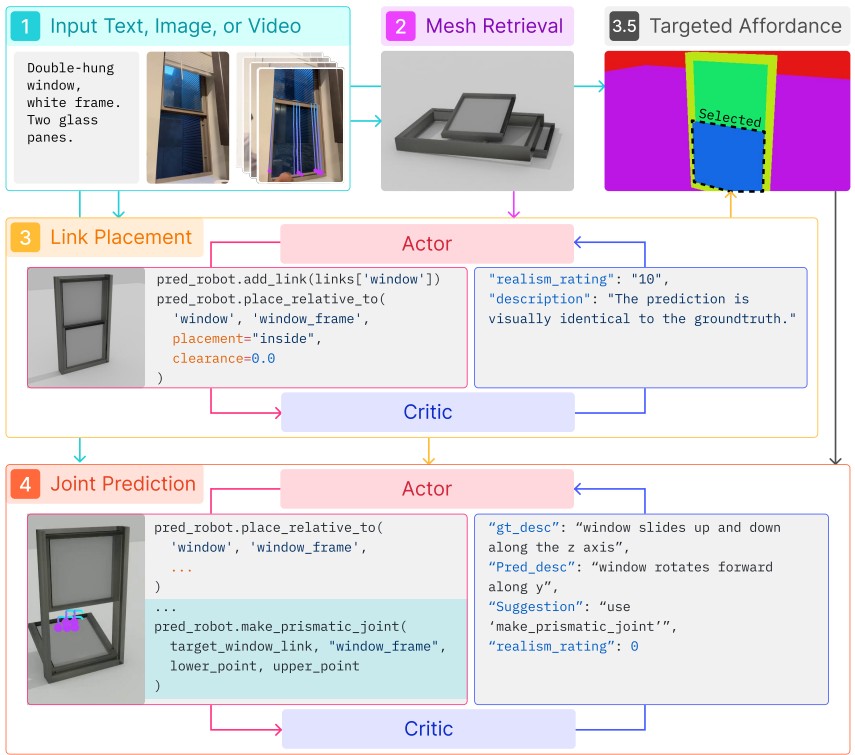

Figure 2: **Method Overview.** Given a text, image or video input, ARTICULATE-ANYTHING operates in three stages: (1) **Mesh Retrieval** (Sec. 4.1) retrieves a mesh for each object part from a 3D asset library, (2) **Link Placement** (Sec. 4.2) places the parts together, and (3) **Joint Prediction** (Sec. 4.3) predicts the allowed kinematic movements between parts. Optionally, instead of generating all possible kinematic joints, we can target a specific joint from the input video (Sec. 4.4). The link placement and joint prediction systems consist of an actor and a critic, which are VLMs working together. The actor proposes solutions, and the critic examines those solutions and gives feedback.

along the global z-axis for a door requires forward kinematics calculations down the URDF tree to translate it to the relative frame. Even seemingly simple tasks, like positioning a toilet seat above the bowl, can be complicated due to non-trivial mesh geometries. However, this can easily be remedied by performing iterative collision checks in a physics engine.

## 4.1 MESH RETRIEVAL

We employ two separate mechanisms for reconstructing meshes from visual (i.e., image or video) and text inputs. An overview of these processes is provided in Fig. 3.

**Visual Inputs.** For visual inputs, we perform mesh retrieval by matching the input to a 3D object from the PartNet-Mobility dataset (Mo et al., 2018). First, a VLM is instructed to detect an object of interest from the input. Since there might be multiple objects in the frame, when a video is provided, the VLM is instructed to take advantage of motion cues for more accurate identification. Then, to reduce the search space, we identify the top-$k$ most similar object categories in the library using CLIP similarity (Radford et al., 2021). Lastly, a VLM agent `Hierarchical ObjectSelector` is tasked with selecting a candidate among several simulated objects based on their visual similarity to the ground truth. This visual similarity estimate is a proxy for the $\mathcal{L}_{\text{mesh}}$ loss described in Sec. 3. To manage the potentially large candidate pool, we adopt a divide-and-conquer tournament approach. Specifically, we recursively select the best candidate among a batch of `max_num_images` images until there is only one candidate left.

**Text Inputs.** For text prompts, we employ a multi-step process using large language model (LLM) agents. First, a `TaskSpecifier` densifies the potentially sparse prompt (please see Fig. 3 (b) for an example). Then, a `LayoutPlanner` is instructed to describe each object part along with its dimensions. Finally, mesh retrieval is performed for each part by matching the object parts' CLIP language embeddings to the mesh descriptions in the PartNet-Mobility dataset via cosine similarity.

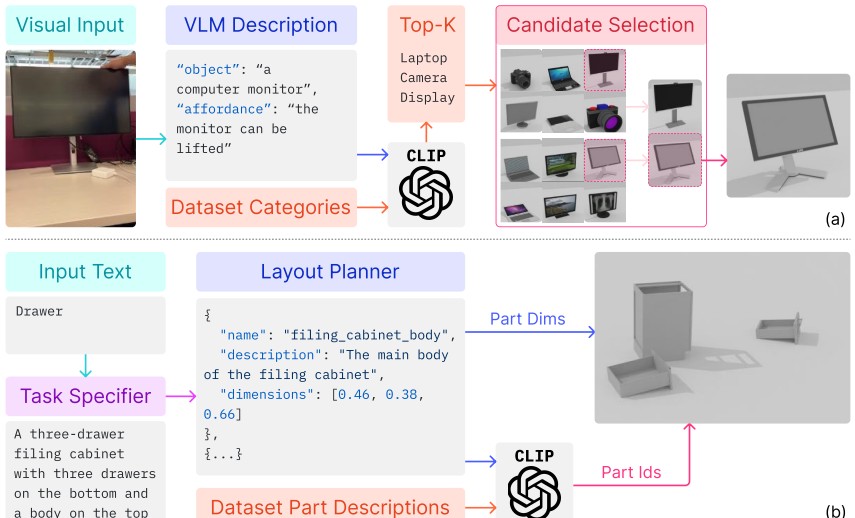

Figure 3: **Mesh retrieval.** The top and bottom diagrams provide overviews for reconstructing visual (i.e., image or video) and text inputs, respectively. For visual input, we match the ground-truth object to a template object in the library using an efficient divide-and-conquer retrieval mechanism. For text input, we first prompt an LLM to predict the different object parts and their dimensions. Then, we retrieve a mesh for each part using precomputed CLIP embeddings and subsequently scale the meshes to specifications. More details in Sec. 4.1.

Since the original mesh descriptions from the dataset are coarse, we compute our own annotations by prompting a VLM to describe each mesh (e.g., material, shape, function) given its image. The retrieved meshes are finally rescaled to fit the specified dimensions.

## 4.2 LINK PLACEMENT

The link placement system can handle text inputs containing the semantic names of each part, along with an optional input image. For videos, we extract the first frame as the image input. The system consists of an actor and a critic. The link actor is responsible for placing links together in the 3D space. In the API, placement is done by aligning aligning the child and parent's link centers along an axis and perform collision checks to ensure tight placement without intersection. When processing visual inputs, a critic is also employed to verify the actor's solution. The critic is prompted to describe any visual dissimilarity between the input image and the predicted 3D model rendered in simulation, pinpoint the source of error in the actor's code, and give a `realism_rating` between 0 and 10. This realism rating serves as a proxy for the $\mathcal{L}_{\text{link}}$ loss. The actor is instructed to take the feedback into account and adjust its Python code accordingly. This process terminates when the rating exceeds a threshold of 5. Figure 4 (Left) provides an illustrative example of this process.

## 4.3 JOINT PREDICTION

Similar to link placement, the joint prediction system accepts inputs in the form of text containing the Python code for link placement, as well as an optional input image or video. The actor then extends the Python code to include kinematic joint prediction between parts. For video inputs, a critic is also utilized to verify the actor's solution. The critic is instructed to compare the input against a video of the predicted joint moving in simulation and examine the source code to identify either success or one of 4 failure cases. A `realism_rating` between 0 and 10, serving as a proxy for the $\mathcal{L}_{\text{joint}}$ loss, is also used to categorize the most egregious error (incorrect joint type) to the least (incorrect joint limit). As before, the actor is instructed to rewrite its code, taking the critic's feedback into account as long as the realism rating falls below a threshold of 5. See an example in Fig. 4 (Right).

## 4.4 TARGETED AFFORDANCE EXTRACTION

Optionally, instead of generating all possible kinematic joints, we can target a specific joint from the input video. This can be useful to reduce the number of input tokens to VLMs, thereby lowering the cost or making automatic debugging of predictions easier. To achieve this, we prompt a VLM agent `targeted_affordance_extractor` to determine which child link should be annotated with a

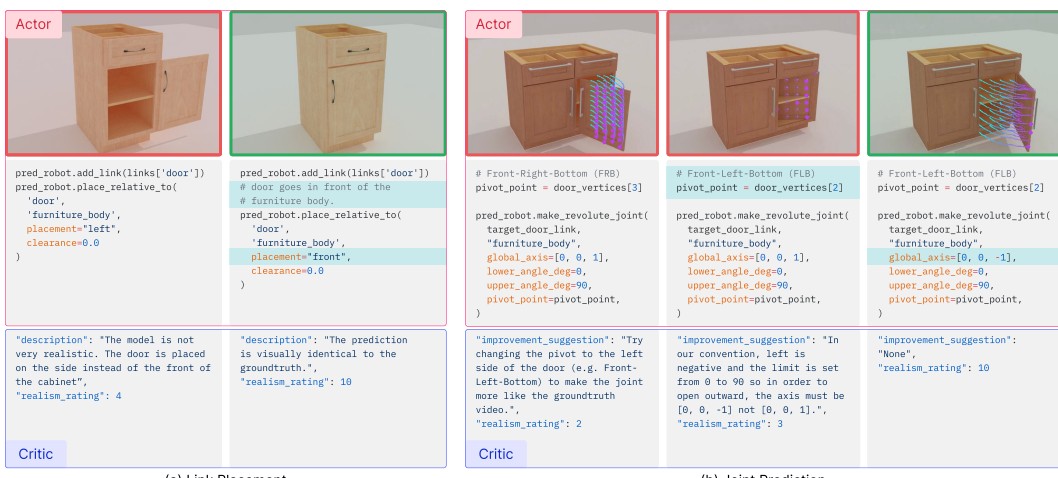

Figure 4: Both link placement and joint prediction systems consist of an actor and a critic. The actor produces Python code, which is automatically compiled into URDFs and rendered in simulation. Source code, predicted, and input modalities (images for link and videos for joint) are given to the critic for evaluation. The synergy between the actor and critic enables self-correction of errors (**red border**) and successful articulation (**green border**).

kinematic joint given the input video and a segmented image of the simulated asset, where each part is distinctively colored.

# 5 EXPERIMENTS

**Datasets:** We use the Partnet-Mobility dataset (Mo et al., 2018) which includes human annotations for ∼2.3K objects, ∼1.9K revolute joints, and ∼7.6K prismatic joints.

**Tasks:** Given a ground-truth articulated URDF from PartNet-Mobility, we mask out all link and joint information. The objective is to reconstruct the masked URDF. Solving this objective includes two primary tasks: (1) link placement—determine the spatial arrangement of object parts, and (2) joint prediction—infer the movement capabilities between parts.

**Evaluation Metrics:** We assess performance using success rates for each task.

1. **Link placement:** Success is determined by the pose difference between predicted and ground-truth links falling below a small threshold.

2. **Joint prediction:** We evaluate differences in joint type, axis, origin, and limit. Success requires all criteria to be within a small threshold.

The position threshold is set to 50mm and the angular threshold to 0.25 radian (∼ 14.3 degree). When evaluating ARTICULATE-ANYTHING, if an object's link placement is incorrect, all of its joints are also counted as incorrect. More mathematical details and some visualizations of the different joint errors are available in Appendix A.1.

**Baselines**: We compare against two prior state-of-the-art methods: URDFormer (Chen et al., 2024) and Real2Code (Mandi et al., 2024). Both methods were trained or fine-tuned on five object categories in the PartNet-Mobility dataset. We evaluate the performance of these five (in-distribution) and the remaining 41 (out-of-distribution) classes. Implementation details are provided in Appendix A.4.

**Implementation Details for ARTICULATE-ANYTHING:** Our system employs Google's Gemini Flash-1.5 (Team et al., 2023) as the vision-language model (VLM). Physics computations are performed using PyBullet (Coumans & Bai, 2016), while rendering is done in Sapien (Xiang et al., 2020). Motion traces in videos are annotated using CoTracker (Karaev et al., 2023) for visualization. We use few-shot prompting with around 20 in-context examples.

## 5.1 HOW WELL DOES ARTICULATE-ANYTHING PERFORM VERSUS PRIOR WORK?

We benchmark ARTICULATE-ANYTHING against two prior works in automatic articulation. In this experiment, ARTICULATE-ANYTHING is fed input videos. The key differences between ARTICULATE-

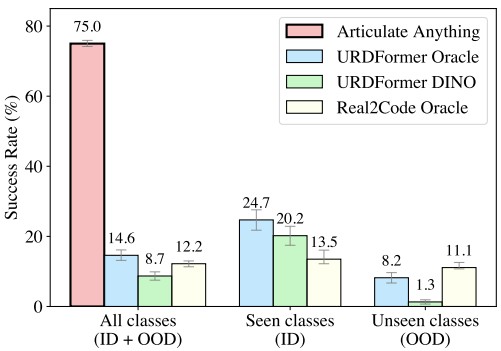

Figure 5: **Comparison against the baselines.** Our approach significantly outperforms all baselines in the joint prediction task. We use few-shot prompting and make no distinction between ID and OOD classes, so we only report results for all classes. 95% confidence intervals are included.

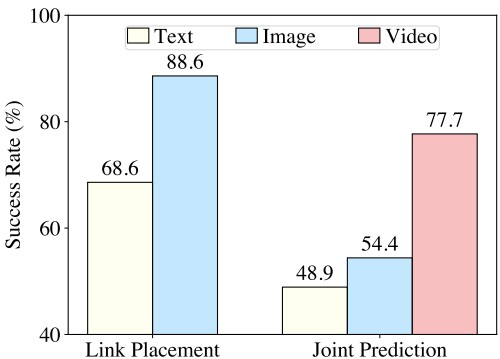

Figure 6: **Input modality ablation.** Performance on articulation tasks improves with more grounded modalities. Videos are only used for joint prediction. Input videos provided to the link placement system are automatically transformed into image inputs by extracting the first frames.

ANYTHING and prior work include (1) our system operates on high-level Python abstraction instead of low-level inputs, (2) our system can handle more grounded instruction inputs, including videos, and (3) our actor-critic system allows iterative refinement on more difficult objects. In contrast, URDFormer directly predicts part position (discretized) coordinates, and Real2code operates on oriented bounding box coordinate inputs. Prior works are also limited to simplified inputs as they cannot handle videos. This also means that their articulation must be done in an open loop.

Our innovations allow ARTICULATE-ANYTHING to achieve superior performance as demonstrated in Fig. 5. In Appendix A.6, table 1 reveals the raw joint prediction errors behind the success rate of Fig. 5, including errors of joint type, joint origin, and joint axis. The error distribution is visualized via violin plots in Fig. 20. Fig. 8 breaks down the failure reasons for each method. The breakdowns of success rate of our method by object categories are provided in Fig. 21 and 22. We also provide an ablation where our method is given the same impoverished input modality as the baselines in Appendix A.5. Mesh reconstruction comparison is given in Appendix A.7.

Figure 7 compares ARTICULATE-ANYTHING against the baselines on in-the-wild object reconstruction. As before, we select both OOD and ID objects. Our video-based method performs the best while baselines fail on all tasks, susceptible to minor misalignments or segmentation inaccuracies. Notably, leveraging rich video input allows ARTICULATE-ANYTHING to resolve articulation ambiguities. For example, our non-video methods predict the chair leg sliding up and down (for adjusting height), and the window tipping to open. These predictions are actually reasonable given the input images and texts but turn out to be wrong when the ground-truth videos are revealed.

## 5.2 HOW DO DIFFERENT INPUT MODALITIES AFFECT ARTICULATION ACCURACY?

While LLMs trained on internet-scale data possess remarkable common sense abilities, we hypothesize that visual inputs are still crucial for accurate object articulation. For example, consider a simple faucet with a spout and a lever. Where should the lever be placed? In fact, there is no standard answer: there are faucets whose levers are on top and those whose levers are to the side. Joint prediction without videos is similarly difficult (e.g., see Fig. 7). Thus we contend that more grounded input modalities such as videos are crucial in resolving difficult or ambiguous articulation.

In this section, we compare the articulation success rates across three input modalities: `text`, `image`, and `video`. For the `text` modality, we provide only the semantic part names for link placement; during joint prediction, the model receives the Python code for link placement and is tasked with inferring the joints. With the `image` modality, we additionally supply a static image of the ground-truth object in addition to the textual information. For the `video` modality, a video showcasing the object's motion is provided in addition to the text. Note that videos are only used for joint prediction. Input videos provided to the link placement system are automatically turned into images by extracting the first frames.

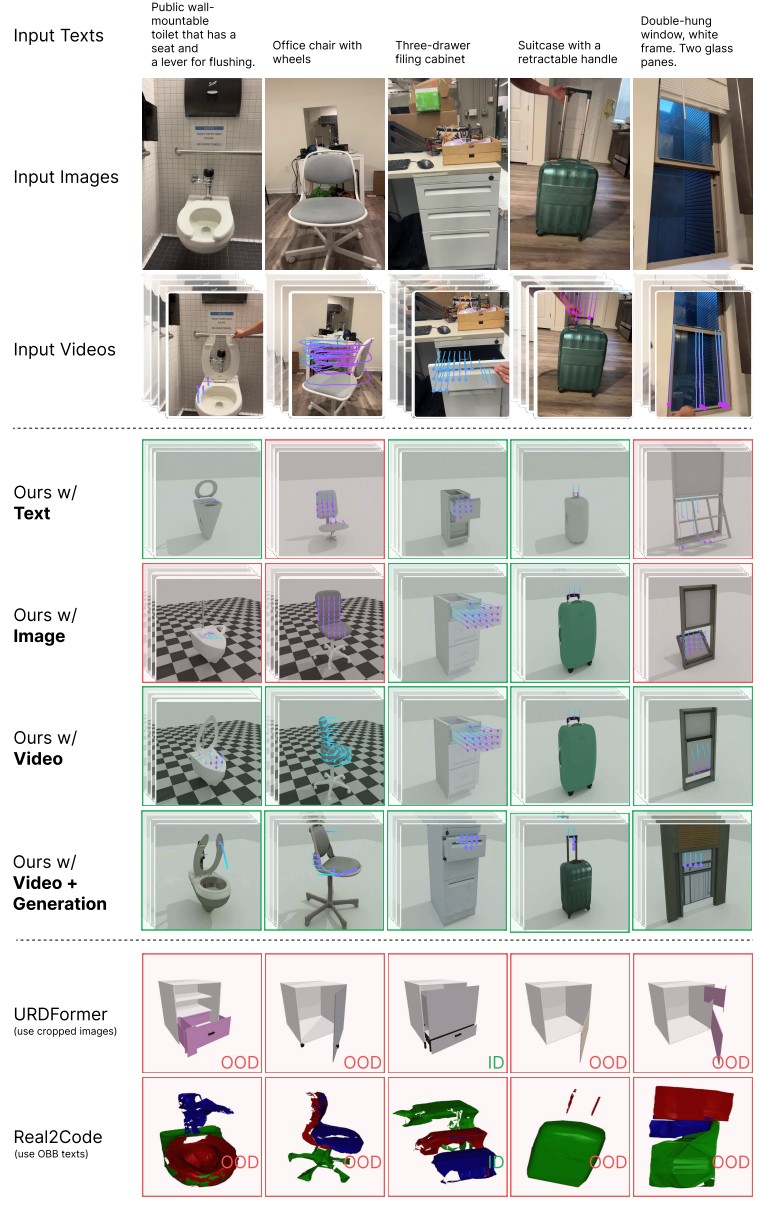

Figure 7: **In-the-wild Reconstruction.** We demonstrate ARTICULATE-ANYTHING's performance input modalities compared to prior works URDFormer and Real2Code. **Green** and **red** borders denote correct and incorrect predictions with respect to the input videos. Our simulated videos are generated by varying joint limits and annotated with Cotracker. **Casual inputs**: Our video-based approach excels with casually captured inputs in cluttered environments while baselines require extensive manual curation (more details in Appendix A.4 Fig. 17 and 18). **Ambiguous articulation**: Interestingly, our video-based method can resolve ambiguities in static inputs (e.g., chair rotation vs. height adjustment, window sliding vs. tipping). **Baseline limitations:** URDFormer consistently predicts drawer-like structures and is sensitive to minor misalignments (e.g., slightly tilted drawers). Real2Code, which uses multi-view images for mesh reconstruction and text-oriented bounding boxes (OBBs) for joint prediction, achieves good global alignment from DUSt3R but produces low-quality 3D segmentation using a finetuned SAM, leading to joint prediction errors. We also included some results using a mesh generation model instead of retrieval to allow for even more customized asset creation with more details in Appendix A.8. Mesh reconstruction quality result is included in Tab. **??** in Appendix A.6. Please visit our website for additional demos: https://articulate-anything.github.io/.

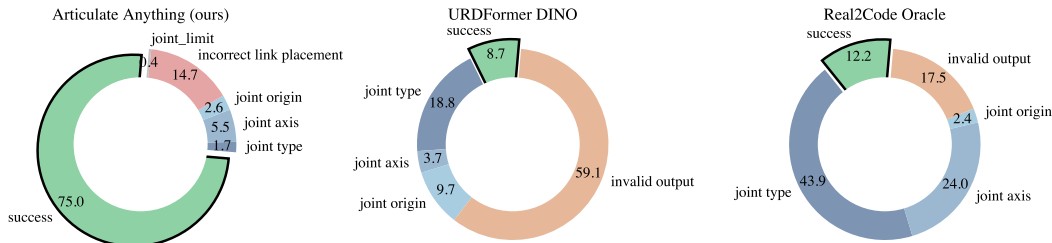

Figure 8: **Breakdown of failure percentages in all classes.** In ARTICULATE-ANYTHING, incorrect link placement leads to all predicted joints being marked incorrect. For baselines, 59.1% of URD-Former's and 17.5% of Real2Code's outputs are invalid, containing cyclic structures, repeated links, or syntax errors. Details are in Appendix A.1.

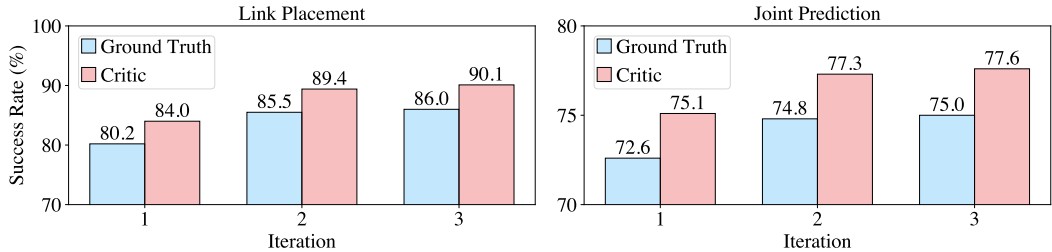

Figure 9: **Iterative Improvement.** ARTICULATE-ANYTHING employs a **critic** agent for self-evaluation and refinement over subsequent iterations. We gain around $5.8\%$ and $2.4\%$ improvement on link placement and joint prediction, respectively. While the **ground-truth** success (based on human annotations) is not available to the agent, our **critic**'s evaluation closely correlates with ground-truth. The confusion matrix is provided in Appendix Fig. 15.

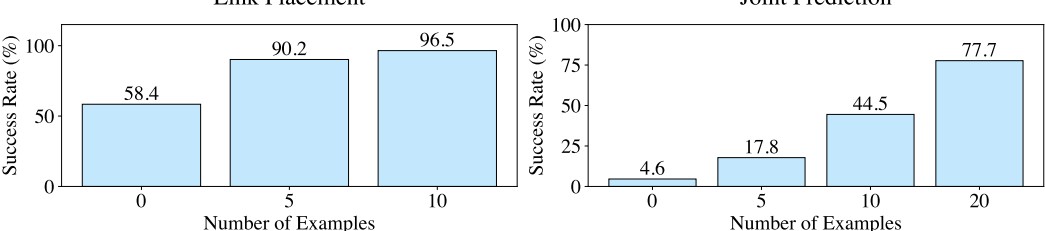

Figure 10: **In-context learning.** ARTICULATE-ANYTHING improves with the number of prompting examples, demonstrating in-context learning. The zero-shot performance (0 example) is included.

We conduct this ablation study on the `Faucet` object category for link placement and `StorageFurniture` for joint prediction. We run an actor-only system for one iteration without the critic to isolate the effect of input modalities. Results in Fig. 6 demonstrate that richer input modalities consistently improve success rates, underscoring the importance of visual information in articulation tasks.

### 5.3 CAN ARTICULATE-ANYTHING EVALUATE AND REFINE ITS OWN PREDICTIONS?

We show quantitatively the iterative improvement capability of articulate-anything in Fig. 9. The critic success rate is computed as the percentage of predictions that are deemed successful by our VLM critic while the ground-truth success rate is computed from human annotations. Via iterative refinement, we gain around $5.8\%$ and $2.4\%$ improvement in accuracy for link placement and joint prediction, respectively. The critic evaluation shows a very high correlation with the ground truth, although it tends to overestimate success (please see Appendix Fig. 15 for the confusion matrix).

### 5.4 HOW DO THE NUMBER OF IN-CONTEXT EXAMPLES AND THE CHOICE OF VISION-LANGUAGE MODELS AFFECT OUR PERFORMANCE?

Frontier VLMs are known to benefit from in-context examples. Furthermore, the performance of these models can vary across architectures. In this section, we investigate how these factors affect our system. We conduct this ablation study on the `StorageFurniture` category using an actor-only system from input videos. Figure 10 shows ARTICULATE-ANYTHING's ability to perform in-context

learning, improving the success rate with increasing number of prompting examples. Figure 11 shows ARTICULATE-ANYTHING's robustness to different choices of leading VLMs, maintaining high success rates throughout.

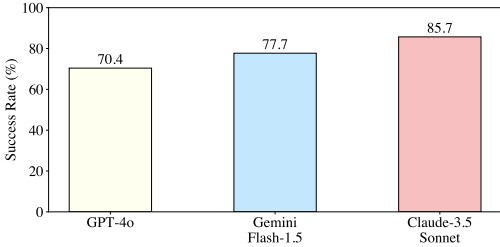

Figure 11: **Base VLMs.** Our method is robust to the choice of VLMs, maintaining high success rates throughout.

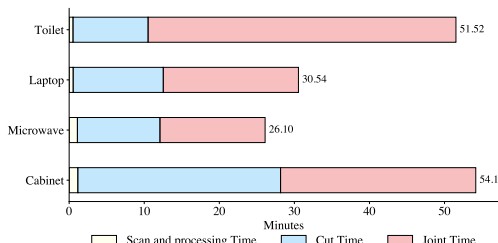

Figure 12: **Human annotation time.** Our method on average saves 40.58 minutes in human manual labor per task.

## 6 AN APPLICATION IN ROBOTICS

A 3D model without articulation can only afford trivial interaction such as pick and place. In this section, we show that generated assets from ARTICULATE-ANYTHING can be leveraged to train reinforcement learning (RL) policies in simulation to perform four finer-grained manipulation tasks. We train these policies using PPO (Schulman et al., 2017) in Robosuite (Zhu et al., 2020; Todorov et al., 2012), with details given in Appendix A.3. These policies are then executed on a real Franka arm. For comparison, we also use a human expert to manually annotate the assets via the RialTo (Torne et al. (2024)) GUI, which had been explicitly designed to facilitate the scanning and annotation process. Both policies trained via ARTICULATE-ANYTHING's and human-annotated assets achieve 100% success rate in the real world. However, ARTICULATE-ANYTHING would have saved an average of 40.58 minutes in human labor per task as shown in Fig. 12.

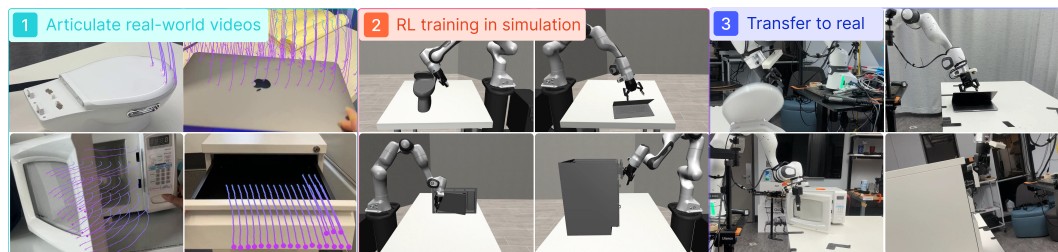

Figure 13: **Robotic Application:** ARTICULATE-ANYTHING can automatically generate assets given in-the-wild input videos. We then train manipulation policies for four tasks in simulation via RL. The policies are deployed back on a real Franka Panda arm, successfully completing all tasks. Please visit our website for the video demonstration: https://articulate-anything.github.io/.

## 7 CONCLUSION AND LIMITATION

In this paper, we have presented ARTICULATE-ANYTHING, a novel vision-language agentic system capable of accurately articulating diverse objects from various input modalities. By reimagining articulation as a program synthesis problem and leveraging the power of VLMs within a self-improving actor-critic framework, ARTICULATE-ANYTHING automates a process previously constrained by intensive human labor and expertise. We hope that our work contributes to bridging the gap between the digital and physical worlds, enabling 3D creators to focus on artistic vision rather than tedious tasks, and paving the way for richer simulated environments that can massively scale up robot learning and interaction. Our current approach relies on mesh retrieval to ensure high-quality 3D assets and focus on the problem of automatic articulation. Integrating ARTICULATE-ANYTHING with an upstream foundation mesh reconstruction model could allow users to generate more customized assets, thereby broadening the system's generality. Some preliminary results on this direction is included in Appendix A.8.

ACKNOWLEDGEMENT

This research was partially supported by the Army Research Office under MURI award W911NF201-0080, the DARPA Triage Challenge under award HR00112420305, DARPA TIA-MAT HR00112490421, Institute for Translational Medicine and Therapeutics and National Center for Advancing Translational Sciences of the National Institutes of Health under Award Number UL1TR001878, the National Science Foundation under NSF CAREER 2239301, NSF 2331783, and the University of Pennsylvania ASSET center. Any opinions, findings, and conclusion or recommendations expressed in this material are those of the authors and do not necessarily reflect the view of DARPA, the Army, the NIH, or the US government.

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

## A   APPENDIX

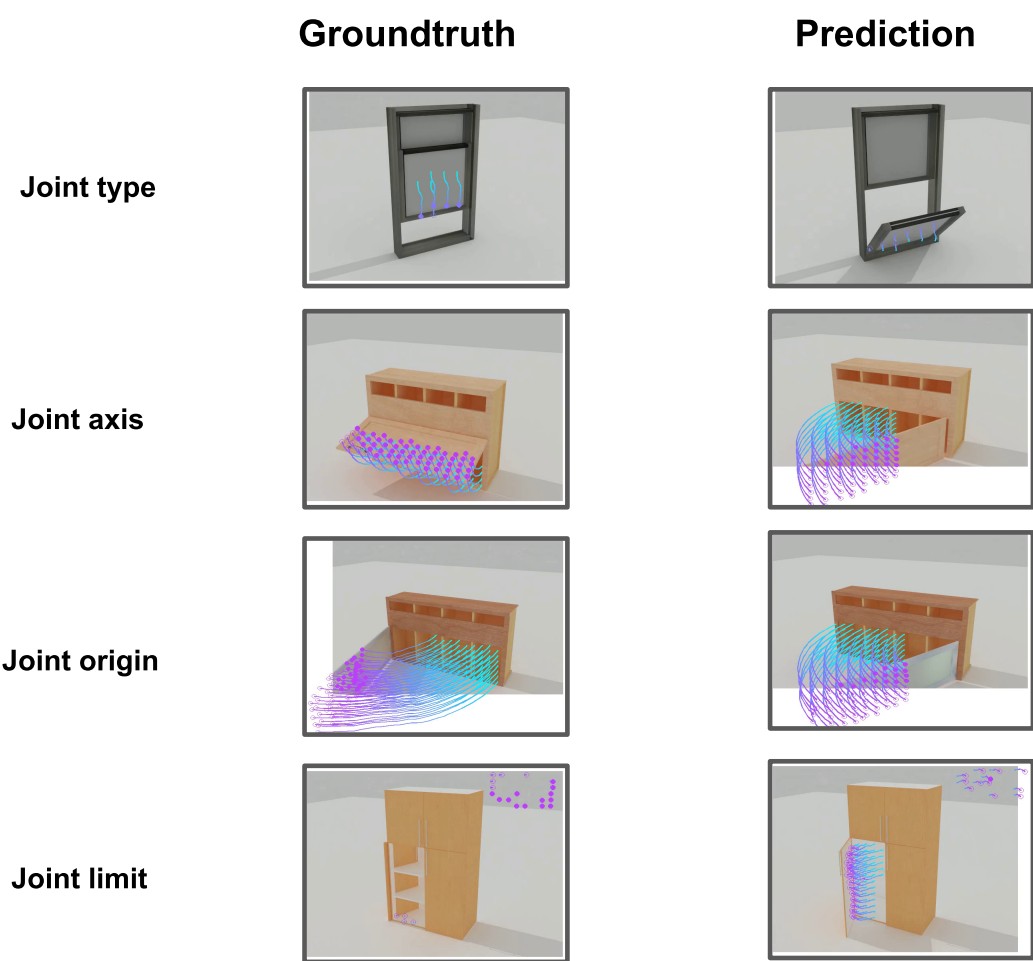

Figure 14: **Joint prediction failure visualization.** We visualize different types of joint failures, ranging from the most egregious, joint type, to the least, joint limit.

### A.1   LINK AND JOINT PREDICTION ERROR COMPUTATION

#### A.1.1   LINK ERROR

A link is represented by a position $\mathbf{x} \in \mathbb{R}^3$ and quaternion orientation $q \in \mathbb{H}$, where $\mathbb{H}$ is the space of unit quaternions. Given a predicted link state $(\mathbf{x}_p, q_p)$ and ground truth state $(\mathbf{x}_g, q_g)$, we define two error metrics:

**Position Error:** The Euclidean distance between predicted and ground truth positions:

$$e_{\text{pos}} = |\mathbf{x}_p - \mathbf{x}_g|_2 \tag{4}$$

**Orientation Error:** The geodesic distance on the unit sphere, computed as:

$$e_{\text{orient}} = 2 \arccos(|q_p \cdot q_g|) \tag{5}$$

where $\cdot$ denotes the quaternion dot product. This metric represents the smallest rotation angle between the predicted and ground truth orientations. The total link error is an average of these components.

```
                                    </>        Code 1: Joint failure attribution.                    </>

def compute_joint_diff_score(
    joint_diff,
    angular_cutoff,
    euclidean_cutoff,
    angular_limit_cutoff,
    limit_cutoff,
):
    """
    return outcome (i.e., `success` or error failure reason)
    """
    if joint_diff["joint_type"] != 1:
        # joint type is not predicted correctly
        return "joint_type"
    elif joint_diff["joint_axis"] > angular_cutoff:
        return "joint_axis"
    elif joint_diff["joint_origin"] > euclidean_cutoff:
        return "joint_origin"
    elif joint_diff["joint_limit"] > limit_cutoff:
        return "joint_limit"
    return "success"
```

### A.1.2 JOINT ERROR

Joint error is computed by comparing several components of the predicted joint state against the ground truth. Let $J_p$ and $J_g$ denote the predicted and ground truth joint states, respectively. The joint error components, from most to least egregious, are as follows:

1. **Joint Type Error:** A binary measure indicating whether the predicted joint type matches the ground truth:

$$e_{\text{type}} = \begin{cases} 0 & \text{if type}(J_p) = \text{type}(J_g) \\ 1 & \text{otherwise} \end{cases} \tag{6}$$

2. **Joint Axis Error:** The angle between predicted and ground truth joint axes:

$$e_{\text{axis}} = \min\left( \arccos\left( \frac{\mathbf{a}_p \cdot \mathbf{a}_g}{|\mathbf{a}_p|_2 |\mathbf{a}_g|_2} \right), \arccos\left( \frac{-\mathbf{a}_p \cdot \mathbf{a}_g}{|\mathbf{a}_p|_2 |\mathbf{a}_g|_2} \right) \right) \tag{7}$$

where $\mathbf{a} \in \mathbb{R}$ is the axis of rotation for revolute joints or of translation for prismatic joints, and $e_{\text{axis}} \in [0, \pi]$.

3. **Joint Origin Error:** For revolute joints, we compute the shortest distance between the joint axes using cross product:

$$e_{\text{origin\_pos}} = \frac{|\mathbf{p} \cdot (\mathbf{a}_p \times \mathbf{a}_g)|}{|\mathbf{a}_p \times \mathbf{a}_g|} \tag{8}$$

where $\mathbf{p} = \mathbf{x}_p - \mathbf{x}_g$ is the difference between predicted and ground truth origins, and $\mathbf{a}_p, \mathbf{a}_g$ are the predicted and ground truth joint axes. For prismatic joints, we use the Euclidean distance:

$$e_{\text{origin\_pos}} = |\mathbf{x}_p - \mathbf{x}_g|_2 \tag{9}$$

4. **Joint Limit Error:** Comprises two sub-components:
   (a) *Motion Range Difference:*

   $$e_{\text{limit\_range}} = |\mathbf{m}_p - \mathbf{m}_g|_2 \tag{10}$$

   where $\mathbf{m}_i = \mathbf{a}_i(u_i - l_i)$, with $u_i$ and $l_i$ being the upper and lower joint limits.
   (b) *Motion Direction Difference:*

   $$e_{\text{limit\_dir}} = 1 - \frac{\mathbf{m}_p \cdot \mathbf{m}_g}{|\mathbf{m}_p|_2 |\mathbf{m}_g|_2} \tag{11}$$

   This metric ranges from 0 (identical direction) to 2 (opposite direction), with 1 indicating perpendicular directions.

A joint prediction succeeds when all of the joint component predictions are within a tolerance. Otherwise, a joint failure is attributed in a natural order as in Code 1 where is the most egregious error – joint type – is detected first and the least – joint limit – last.

A visualization of different joint failure types is given in Fig. 14.

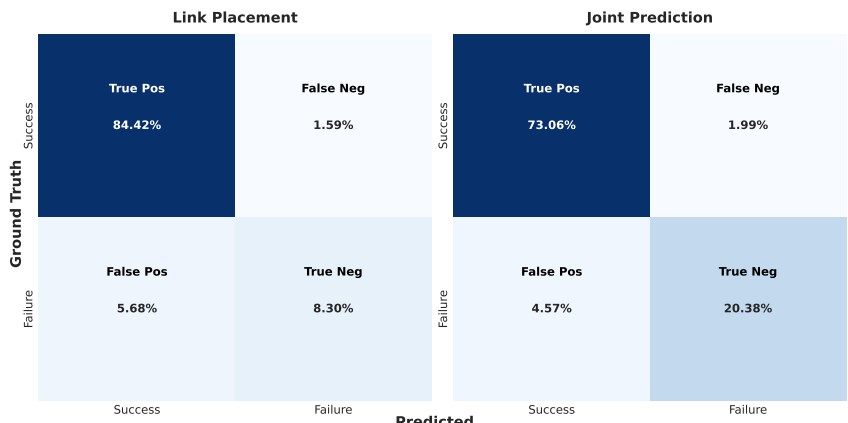

Figure 15: **Critic-groundtruth confusion matrix.** Our visual critic is highly correlated with the ground-truth. The largest disagreement comes from false positive case i.e., the critic falsely declares an incorrect articulation as correct. These cases include difficult-to-notice errors.

## A.2 Critic-groundtruth Agreement

Fig. 15 visualizes the confusion matrices between the ground-truth and our own critic predictions of success in two articulation tasks. Our visual critic is highly correlated with the groundtruth. The largest disagreement comes from false positive case i.e., the critic falsely declares an incorrect articulation as correct. These cases include difficult-to-notice errors.

## A.3 Robotic Training Details

We train a Franka arm to perform four robotic manipulation tasks in the Robosuite simulator using PPO and our generated assets.The policy outputs joint and gripper positions. We train policies over 3 random seeds per task for 2 million environment steps using PPO in Stable-Baselines3 library Raffin et al. (2021). We randomize physics (friction, damping, frictionloss ect), objects' scales and poses to obtain robust policies.

## A.4 Baselines

URDFormer requires a bounding box for each object part in the input image, which can be obtained via a fine-tuned Grounding DINO (Liu et al., 2023c) or an oracle using ground-truth boxes from a physics engine. Real2Code requires oriented object bounding boxes (OBBs) as input texts to query a LLM. On PartNet-Mobility, these were obtained using oracle RGB-D images and ground-truth segmentation masks from Blender.

### A.4.1 Casual Inputs

Our video-based approach excels with casually captured inputs in cluttered environments while baselines require extensive manual curation: we adjusted DINO bounding boxes for URDFormer and carefully curated foreground segmentation and other hyper-parameters for Real2code. On our project website, we show a diversity of object categories casually captured on an iPhone. The inputs have different view angles and are sometimes inadvertently titled. This kind of inputs presents great difficult for the baselines. For example, Fig. 18 shows the manual adjustment needed for URDFormer. Fig. 17 shows the manually curated foreground segmentation mask and other tuned hyper-parameters to clean up object-part masks and point clouds for real2code.

### A.4.2 Reproducing Real2code

Real2code LLM Training Real2code has not released their LLM model checkpoint nor training code. We therefore have done our best to reproduce their LLM model based on their paper description and other code. We obtain the LLM training dataset using their preprocessing code, and finetune

```
[INST] You are an AI assistant trained to understand 3D scenes and object
relationships. Given the following Oriented Bounding Box (OBB) information, your
task is to generate a list of child joints that describes the articulations between
object parts.

OBB Information:{...}

Generate a list of child joints. Each joint should be described by a dictionary
with the following keys:
- box: The ID of the child bounding box
- type: The joint type ('hinge' for revolute joints, 'slide' for prismatic joints)
- idx: The rotation axis index (0 for x-axis, 1 for y-axis, 2 for z-axis)
- edge: Edge coordinates on the OBB, for example [1, -1]
- sign: Direction of the joint (+1 or -1)

IMPORTANT: Your response must contain ONLY the child_joints list, exactly
as shown below, with no additional text before or after:
child_joints = [
dict(box=[child OBB ID], type=[joint type], idx=[rotation axis index], edge=[edge
coordinates], sign=[direction]),
# Additional joints as needed ]

Generate the child_joints list: [/INST]
```

Figure 16: **Our instruction prompt for real2code reproduction.** Neither training code nor model checkpoint were available from the original work's Github.

the CodeLLama-7B-Instruct model Roziere et al. (2023) using LoRA Hu et al. (2021) with 4-bit quantization as described in their paper. The prompt for the LLM is given in Fig. 16.

Real2code In-the-wild reconstruction We were not able to reproduce the shape completion model based on the released code. Thus, we instead use alpha surface reconstruction Edelsbrunner et al. (1983) from Open3D Zhou et al. (2018) to reconstruct meshes from point clouds, and manually tune the hyper-parameters. We found that this method produces on-par or higher quality meshes from their meshes that were publicly released.

We manually select query points for SAM to ensure near-perfect foreground segmentation masks, tune hyper-parameters for the 3D segmentation scheme (e.g., the input images, number of query points, minimum point-cloud size et cetera), clean up point-clouds, masks and tune mesh extraction. The kinematic-aware SAM model was finetuned using the code from the baseline's authors.

We manually curate the foreground segmentation masks, remove incorrectly segmented point-clouds, tune hyper-parameters and manually parse their LLM outputs. Fig. 17 visualizes the inputs and intermediate outputs from real2code.

### A.4.3 EVALUATING URDFORMER

URDFormer predicts one link per bounding box, with each non-root link associated with a joint. To evaluate URDFormer's predictions against ground truth, we employ a link matching algorithm that aligns predicted links with ground truth links. Each URDFormer bounding box is matched with a ground-truth link by computing the overlap between the box and link's segmentation mask given by the physics engine. For in-the-wild reconstruction, we manually corrected the object-part bounding boxes as shown in Fig. 18.

| Input Image (one of many multi-view images) | Foreground Seg Mask (one of many) | SAM Output (one of many) | Segmented Point Clouds | Segmented Meshes |

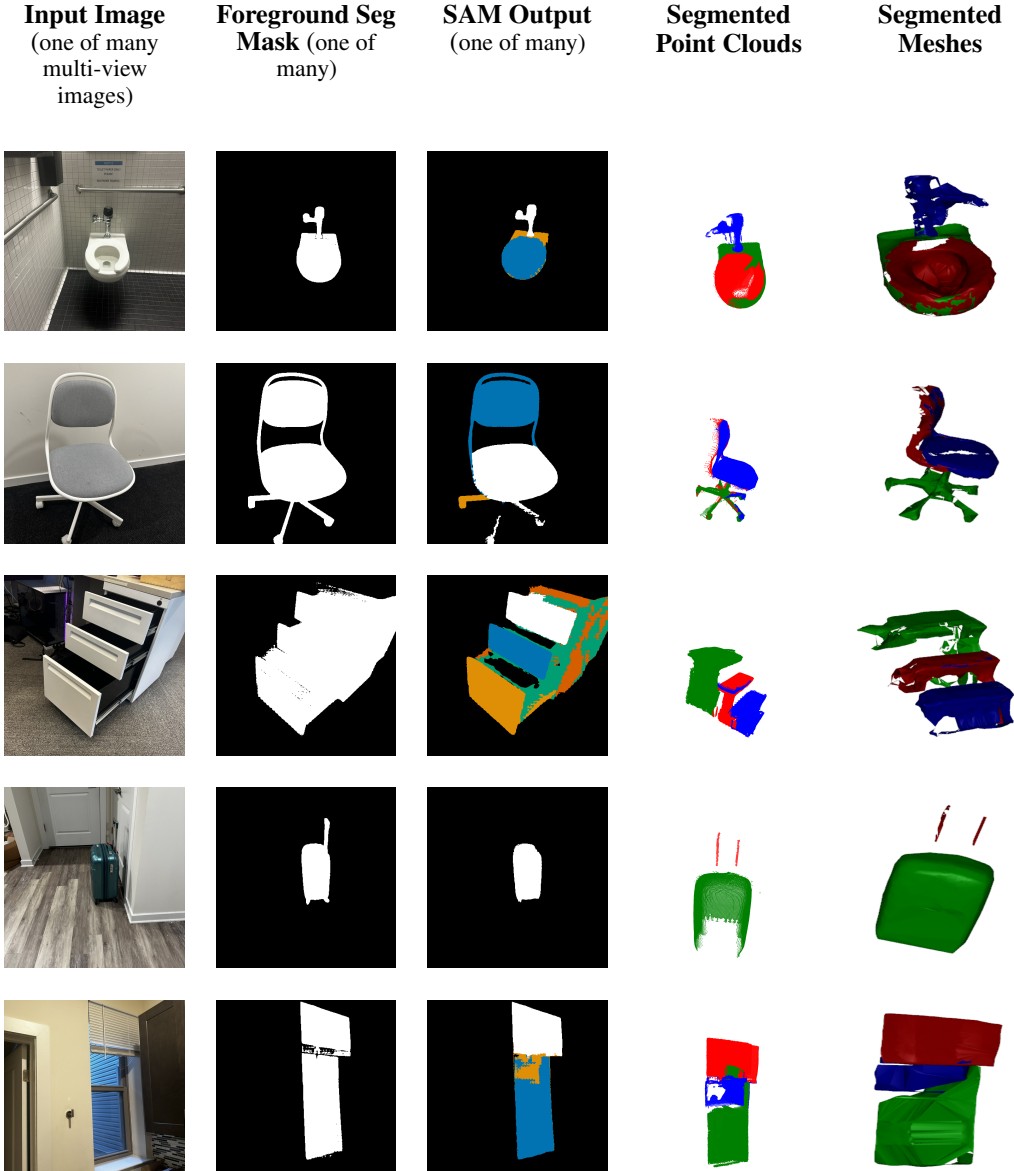

Figure 17: **Real2code manually curated inputs and intermediate outputs.** We used about 3 to 7 input images per object from different views to obtain good global alignment from DUSt3R. Foreground segmentation masks were manually curated by querying the base SAM model. The segmented point-clouds and meshes were curated by tuning various segmentation hyper-parameters (e.g., which input images to use, number of query points, minimum point-cloud size et cetera) and surface reconstruction parameters (i.e., $\alpha$). Incorrectly segmented point clouds were manually removed and spurious points were removed using Open3D's radius outlier removal and only keeping the largest connected component from DBSCAN clustering Ester et al. (1996) per segmented mask. LLM outputs contain incorrect syntax and extra rambling which were parsed manually.

**DINO**                    **Manual**

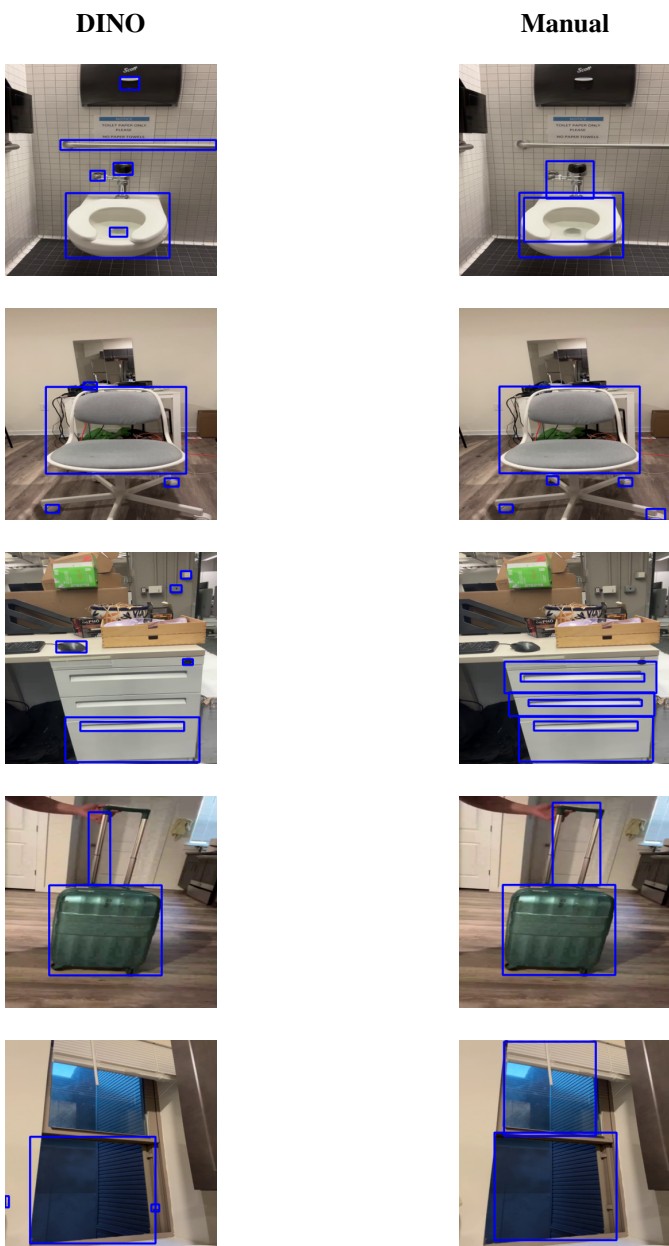

Figure 18: **URDFormer manual correction.** Comparison of bounding boxes from their fine-tuned DINO and our manual correction.

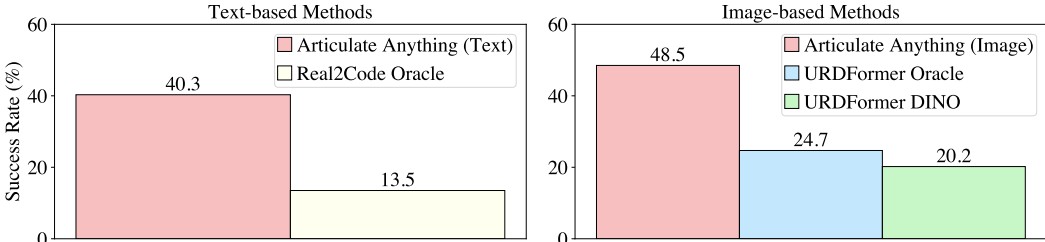

Figure 19: **Comparable Inputs**. We compare ARTICULATE-ANYTHING with two baselines, Real2Code and UDRFormer using the same input modalities. The ablation is done on the corresponding seen classes of the baselines.

Table 1: **Comparison of average joint prediction errors across methods (lower is better).** 'Type' shows the fraction of incorrect joint type predictions, 'Axis' error is measured in radians, and 'Origin' error in meters. ARTICULATE-ANYTHING substantially outperforms all previous works. ARTICULATE-ANYTHING uses few-shot prompting and makes no distinction between ID and OOD classes, so we only report results for All Classes. 95% confidence interval is included for classification error (joint type) and standard deviation for continuous errors (axis and origin).

| Method | All Classes | | | ID Classes | | | OOD Classes | | |
|---|---|---|---|---|---|---|---|---|---|
| | Type ↓ | Axis ↓ | Origin ↓ | Type ↓ | Axis ↓ | Origin ↓ | Type ↓ | Axis ↓ | Origin ↓ |
| Real2Code Oracle | $0.537_{\pm0.014}$ | $1.006_{\pm0.723}$ | $0.294_{\pm0.417}$ | $0.410_{\pm0.029}$ | $1.164_{\pm0.671}$ | $0.344_{\pm0.479}$ | $0.576_{\pm0.016}$ | $0.937_{\pm0.734}$ | $0.272_{\pm0.386}$ |
| URDFormer Oracle | $0.556_{\pm0.025}$ | $0.374_{\pm0.666}$ | $0.581_{\pm0.355}$ | $0.418_{\pm0.036}$ | $0.208_{\pm0.532}$ | $0.609_{\pm0.357}$ | $0.679_{\pm0.032}$ | $0.643_{\pm0.766}$ | $0.513_{\pm0.340}$ |
| URDFormer DINO | $0.460_{\pm0.033}$ | $0.261_{\pm0.583}$ | $0.547_{\pm0.317}$ | $0.288_{\pm0.039}$ | $0.133_{\pm0.437}$ | $0.582_{\pm0.281}$ | $0.722_{\pm0.047}$ | $0.758_{\pm0.782}$ | $0.438_{\pm0.385}$ |
| ARTICULATE-ANYTHING | $\mathbf{0.021_{\pm0.003}}$ | $\mathbf{0.141_{\pm0.441}}$ | $\mathbf{0.200_{\pm0.380}}$ | - | - | - | - | - | - |

Figure 20: **Violin plots of the continuous errors.** Joint type is binary error and is not applicable for this plot. ARTICULATE-ANYTHING has the smallest average origin and axis error as presented in Tab. 1. For joint axis, we notice two distinct modes in the baselines, indicating the errors for in-distribution and OOD data.

## A.5 COMPARABLE INPUT MODALITIES

Fig. 19 compares ARTICULATE-ANYTHING against the baselines using the same input modalities. We achieve a higher accuracy than previous work using the same impoverished modalities by leveraging foundation models' common sense and high-level planning capability.

## A.6 ADDITIONAL STATISTICS

Figures 21 and 22 provide a breakdown of the success rate by each object category for the link placement and joint prediction tasks respectively. We use the camera pose for all tasks. Small object parts or exotic irregular movement tend to pose greater difficulty to ARTICULATE-ANYTHING. Further optimizations such as zooming in with camera might improve performance. Table 1 provides the raw average errors for our method and prior work with statistical confidences included. Figure 20 further visualizes the distribution of errors via violin plots.

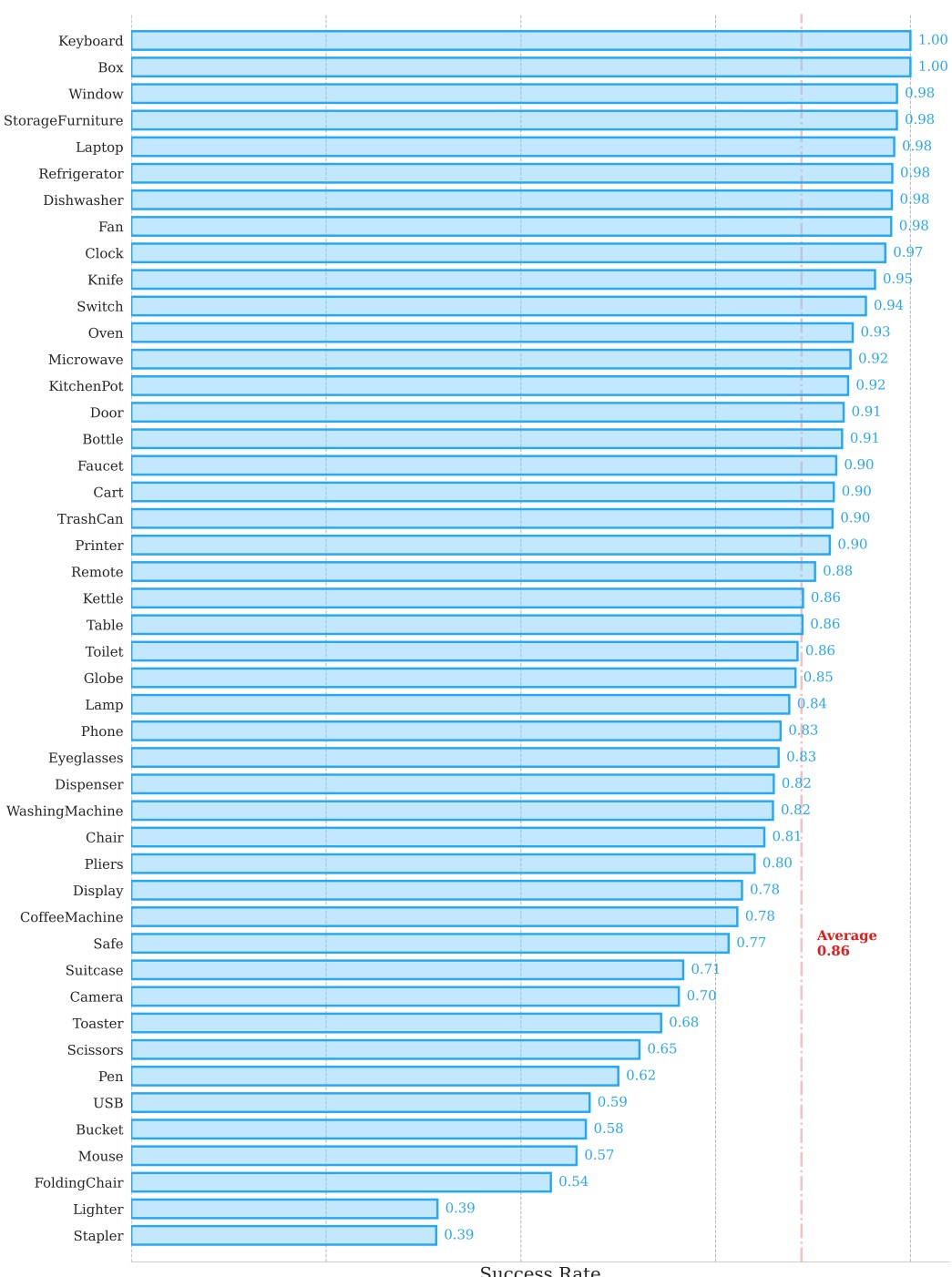

Figure 21: ARTICULATE-ANYTHING's link placement success rate by object categories.

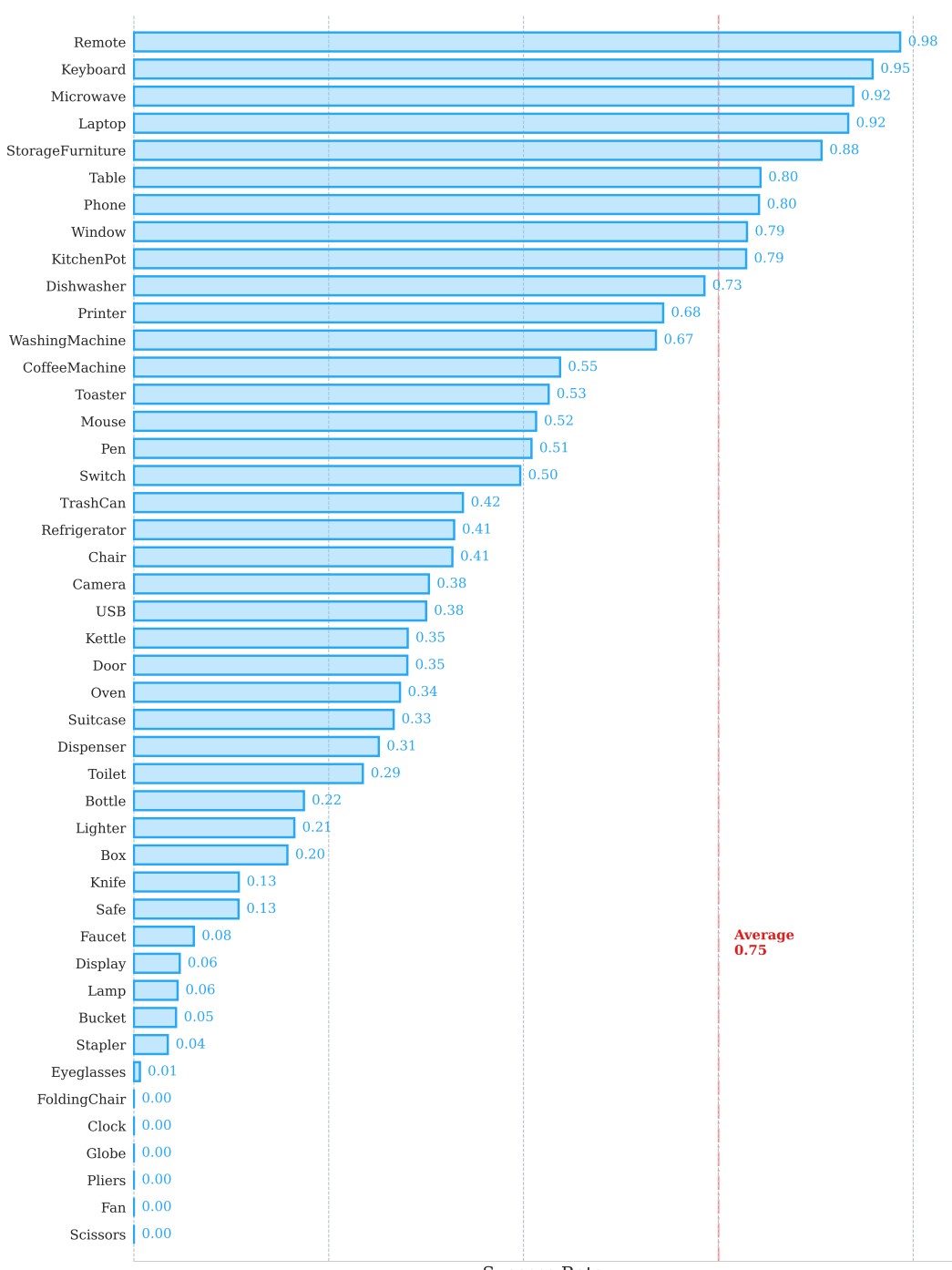

Figure 22: ARTICULATE-ANYTHING's joint prediction success rate by object categories.

## A.7 MESH RECONSTRUCTION

Table 2: **Mesh reconstruction quality**. Chamfer distance is included (lower is better) for different models for in-the-wild results. Best results are **bolded**, second best are underlined. ARTICULATE-ANYTHING integrated with a mesh generation is far better than any other baselines. Even with retrieval, our system is still better than all prior works, including Real2Code which uses DUSt3R to do explicit 3D scene reconstruction.

|  | Toilet | Cabinet | Suitcase | Chair | Avg. |
|---|---|---|---|---|---|
| Ours (Generation) | **0.0637** | **0.0740** | **0.0735** | **0.0698** | **0.0703** |
| Ours (Retrieved) | 0.1133 | 0.1215 | 0.0781 | 0.1210 | 0.1102 |
| Real2Code | 0.1192 | 0.1397 | 0.0877 | 0.0987 | 0.1085 |
| URDFormer | 0.4191 | 0.2164 | 0.1531 | 0.1278 | 0.2291 |

Table 3: Comparison of Chamfer distances across different methods. Lower values indicate better mesh reconstruction quality. Means and standard deviations are included.

| Method | Chamfer distance |
|---|---|
| Articulate-Anything (retrieval) | **0.1007 ± 0.062** |
| Real2Code (Oracle) | 0.229 ± 0.166 |
| URDFormer (Oracle) | 0.429 ± 0.267 |
| URDFormer (DINO) | 0.437 ± 0.217 |

Table 2 includes the mesh reconstruction results for in-the-wild objects in Fig. 7. The ground-truth meshes are captured on a Lidar-equipped iPhone. Window is not included as Lidar does not capture glasses well. ARTICULATE-ANYTHING integrated with a mesh generation is far better than any other baselines. Even with retrieval, our system is still better than all prior works, including Real2Code which uses DUSt3R to do explicit 3D scene reconstruction. Table 3 compares the reconstruction quality on the PartNet-Mobility dataset. Ground-truth RGBD images were used by Real2Code. When evaluating an object, we remove that object from the candidate pool, preventing ARTICULATE-ANYTHING from retrieving the exact same object.

## A.8 GENERATING NEW ASSETS

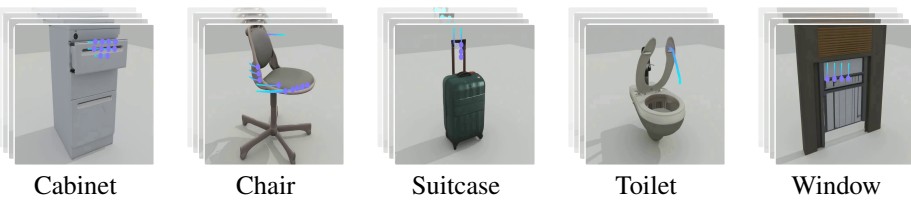

Cabinet          Chair          Suitcase          Toilet          Window

Figure 23: **ARTICULATE-ANYTHING with mesh generation**. ARTICULATE-ANYTHING can be integrated with a mesh generation model to produce high-quality articulated objects. Please see our website for the video demonstrations: https://articulate-anything.github.io/.

Currently, ARTICULATE-ANYTHING employs a mesh retrieval mechanism to utilize existing 3D datasets. Open repositories such as Objaverse (Deitke et al., 2024), which contain over 10 million static objects, offer a rich source of assets that our system can bring to life through articulation. However, to generate even more customized assets, a promising future direction is to leverage large-scale mesh generation models.

In this section, we present preliminary results toward this goal. Given visual inputs to ARTICULATE-ANYTHING —such as videos or images—we first extract a single image of the target object. A mesh generation model is then used to produce a 3D model of the object. 24 compares the mesh quality produced by three models: Rodin (Deemos, 2024), Instant Mesh (Xu et al., 2024), and Stable-Fast-3D (Boss et al., 2024).

We select Rodin for its high-quality output. Subsequently, we use Grounded Segment-Anything (Ren et al., 2024) to obtain a 3D segmentation of the object into parts. Unconditional segmentation is unreliable because object parts may be under- or over-segmented, depending on the task. To address this, we condition the segmentation on the video inputs by instructing a VLM to identify the moving part, which serves as the segmentation target. We then render the 3D model from multiple camera views. For each view, we apply Grounded SAM, which first obtains a bounding box for the object part and then runs SAM to obtain a fine-grained segmentation mask. Figure 25 visualizes the 2D segmentation masks for each view.

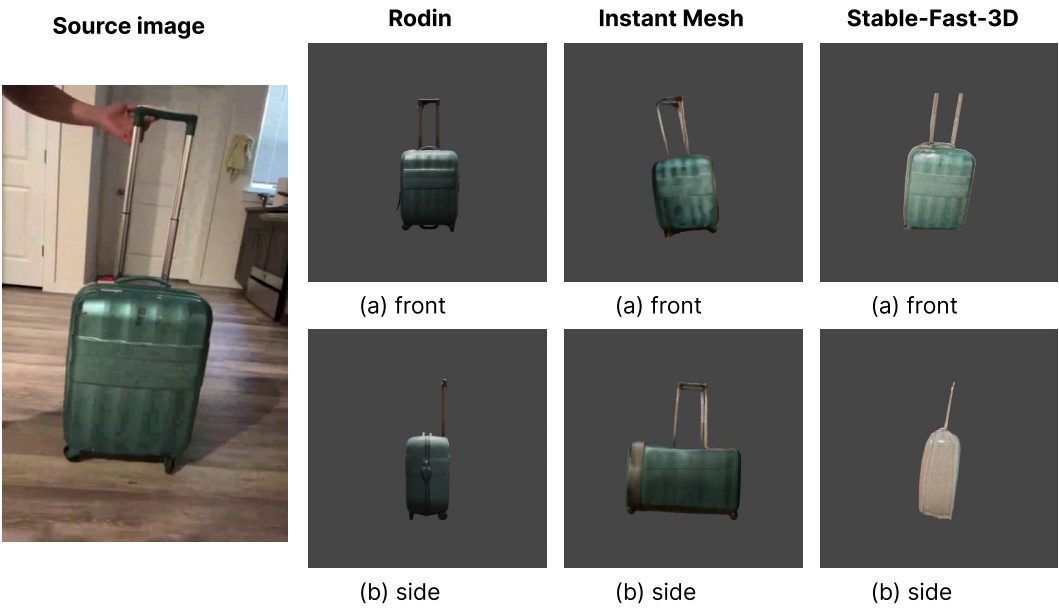

Figure 24: **Mesh generation**. Comparison of 3D mesh generation from a single source image using different models. Each model's output is shown from the front and side views.

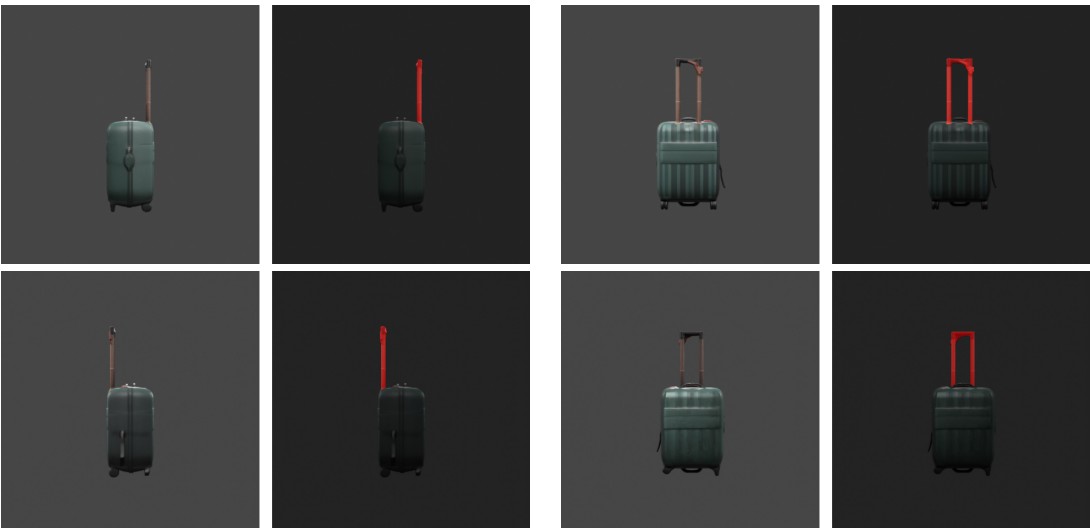

Figure 25: **Grounded SAM**. A 2D segmentation mask is obtained for each view of an object using Grounded SAM. For each view, the input RGB image is shown on the left, and the resulting mask overlaid in red is shown on the right.

The 2D masks are lifted into 3D via projective geometry. For each view, we use the camera parameters and depth maps to project the 3D points into pixels. Then, the 2D segmentation masks are queried to obtain segmentation labels for the 3D points. We merge these segmented 3D points across all

views to obtain a complete 3D segmentation of the object into its semantic parts. The potentially sparse segmented point clouds are used to segment the dense meshes via nearest neighbor matching. With the 3D object now properly segmented, we can apply ARTICULATE-ANYTHING as before. The resulting 3D objects not only possess high material quality that matches the ground truth but also exhibit realistic movement when articulated in simulation. Please refer to our website for the video demonstrations.

