# OpenReview forum: "Articulate-Anything:  Automatic Modeling of Articulated Objects via a Vision-Language Foundation Model"
_ICLR.cc/2025/Conference — ICLR 2025 Poster_

### Official Review · Reviewer_Bhiq · 2024-10-16

**Soundness:** 4
**Presentation:** 3
**Contribution:** 4
**Rating:** 8
**Confidence:** 5

**Summary:**

This work proposes a retrieval method for generating articulated objects in a simulator, taking into account both link and joint placement. It also demonstrates promising results in partner mobility

**Strengths:**

The proposed method for generating digital twins of articulated objects is intriguing and highly effective, significantly enhancing performance during evaluation. The website is comprehensive, offering visualizations and related code.

**Weaknesses:**

Please refer to 'Questions' section

**Questions:**

- Is there any failure case analysis? For example, under what circumstances does the retrieval process select the wrong mesh? It would be helpful to show when retrieval might fail and how challenging those cases are.

- Concerns regarding Section 4.2:
  1) For instance, in Fig. 4a (left example), 'furniture body'  is place to the left relative to the 'door'. Since the body has thickness, why is the door placed on the left side of the front surface rather than the back? As the description only defines 2D spatial relationships, it's unclear how to position the link in 3D space.
  2) How many candidates does the actor's output generate? Can the relation fully cover all possibilities?

- Regarding Section 6's robotic experiment, where do the generated assets come from? Does Articulated Anything input video or images from RoboSuite to generate assets, and then train PPO with the generated assets, instead of directly using the assets in RoboSuite?

---

> ### Author Response · Authors · 2024-11-22
>
> Thank you for your positive review! We have updated our draft to incorporate your feedback.
>
> **When does the retrieval process fail?**
>
> As Fig. 7 shows, the retrieved objects indeed might not fully resemble the real-world objects (for example see the chair). This is quantified by the Chamfer distance in Table 2 (Appendix A.5). Nonetheless, we note that our articulation system is general purpose and can be integrated with any mesh acquisition method. Specifically, we have added new results in Fig. 6, 22, and our [website](https://iclr-2025-4168.github.io/) where meshes generated by a diffusion model fully resemble the real objects.
>
> **Robotic experiment details**
>
> The simulation assets are generated by Articulate-Anything given in-the-wild videos as shown in the updated Fig. 13. We have spent significant effort developing and testing the control stack for our hardware system and have now included additional results showing direct sim2real transfer.  Please see the updated Sec. 6 and [website](https://iclr-2025-4168.github.io/). We further show that compared to human labeling, Articulate-Anything saves, on average, 40.58 minutes of annotation time per task.
>
> **Other questions**
> - Clarification on 3D positioning
> > The link placement works by aligning the child and parent's link centers along an axis and performing collision checks to ensure tight placement without intersection. Thank you for your insightful observation! For the specific Fig. 4(a) case, this is due to the specific way that the PartNet-Mobility mesh is off-setted in their dataset so not all meshes are centered at the origin. In general, more granular positioning such as `placement=left_front` can be implemented in the API. Alternatively, our VLM can be prompted to chain `front` and `left` placements together to achieve the same effect using in-context learning.
>
> - Number of actor candidates
> > The actor generates python code until one code candidate can be run without any compiler error. The maximum number of candidates
> is set to 3. We have found that our system can generate at least a compilable program on the first try in most cases, or on a second try in the worst case.
> - Can the API support all cases?
> > Both the link placement and joint prediction APIs can cover all possible cases without loss of generality. For example, for link placement, we have an optional `clearance` (i.e. offset) parameter that can be added to the placement. Placements can further be chained together by making multiple python calls, thereby allowing for a link to be placed in any position in the 3D space.

---

> > ### Author Response · Authors · 2024-11-25
> >
> > Dear Reviewer Bhiq,
> >
> > Thank you again for your review. As we near the end of the discussion period, we kindly ask that you review our response to your feedback and consider adjusting your score taking into account our revision.
> >
> > Thank you.

---

> > > ### Comment · Reviewer_Bhiq · 2024-11-25
> > >
> > > Thank you for your response, which clarifies the process for me. As a result, I will maintain the rating and am inclined to accept.

---

### Official Review · Reviewer_VbEW · 2024-10-21

**Soundness:** 2
**Presentation:** 3
**Contribution:** 2
**Rating:** 6
**Confidence:** 3

**Summary:**

This paper presents a system designed to automate the articulation of complex 3D objects for applications in AR/VR, animations, and robotics, addressing the challenge of extensive human effort required in traditional methods. It utilizes vision-language models (VLMs) to convert inputs such as text, images, and videos into interactable digital twins for 3D simulators. The system incorporates a mesh retrieval mechanism from existing 3D datasets and actor-critic frameworks to iteratively refine object articulation, achieving robust results.

**Strengths:**

1. The proposed methods for automatically modeling articulated objects address an interesting and meaningful topic, with potential to greatly benefit fields such as robotics.

2. The techniques introduced are well-founded. By leveraging vision-language models (VLMs), which encode valuable information about object affordances, the system effectively determines common articulations for various objects.

3. The paper provides a detailed analysis of failure cases, as shown in Figure 8, highlighting how the proposed pipeline outperforms other methods and identifying areas for further improvement.

4. The paper is well-written and easy to follow, making the concepts accessible to readers.

**Weaknesses:**

1. The proposed methods have only been validated on limited types of articulated objects (i.e., prismatic and revolute). When the types become more complex, such as cylindrical or universal joints, the current pipeline may struggle to handle these cases.

2. The pipeline has only been tested using Google's Gemini VLM. It is recommended to conduct ablation studies to assess whether the inference performance varies with different VLM architectures, in order to evaluate the system's robustness.

**Questions:**

1. Could the authors provide examples illustrating how the performance of link placement and joint prediction evolves over the iterations?

2. Does the number of prompted examples affect the pipeline's performance? It would be helpful if the authors could provide an ablation study on the number of prompted items to determine whether performance varies based on this factor.

**Details Of Ethics Concerns:**

No ethics concerns.

---

> ### Author Response · Authors · 2024-11-22
>
> Thank you for your constructive feedback! Here we addressed your comments:
>
> **Articulated objects in the evaluation**
>
> We use the PartNet-Mobility dataset, which is the largest human-annotated and most widely used dataset for papers in this topic (see [URDFormer](https://arxiv.org/abs/2405.11656), [Real2code](https://arxiv.org/abs/2406.08474), [PARIS](https://arxiv.org/abs/2308.07391), [Ditto](https://arxiv.org/abs/2202.08227) ect.). This dataset includes human annotations for everyday objects, which include the most common joint types: prismatic and revolute. We have not evaluated on more exotic joint types as (1) we are not aware of any dataset that includes them, and (2) to ensure the highest chance of success for our baselines from prior works which have been trained on this exact dataset.
>
> Notably, our evaluation is significantly more comprehensive – we test on all 46 object categories, whereas prior works typically evaluate on only five categories. This extensive evaluation reveals important limitations of existing approaches when handling out-of-distribution data (Fig. 5).
>
>
> **How does different VLM base models or the number of examples affect the performance?**
>
> Thank you for your suggestion! We have added an ablation study on the base VLM model (Fig. 11) and the number of examples (Fig. 10). The results demonstrate our robustness to the choice of VLM, and our ability to do in-context learning.
>
> **Iterative improvement examples**
>
> We have the quantitative results in Fig. 9 and qualitative examples in Fig. 4.

---

> > ### Comment · Reviewer_VbEW · 2024-11-25
> >
> > Thank you for your responses.
> >
> > They were helpful and addressed most of my concerns.
> >
> > However, it seems my last question (Q2) may have been overlooked.

---

> ### Author Response · Authors · 2024-11-25
>
> Dear reviewer VbEW,
>
> Are you referring to the question about the number of prompting examples?
>
> We have addressed this question under our second bolded section. In particular, we have included two additional experiments: robustness to VLM choices (Fig. 11) and the ablation with number of prompting examples (Fig. 10) as you have suggested.

---

> > ### Comment · Reviewer_VbEW · 2024-11-25
> >
> > Thanks for the response.

---

> > > ### Author Response · Authors · 2024-11-25
> > >
> > > Dear Reviewer VbEW,
> > >
> > > Thank you again for your valuable feedback. We have included the two ablation experiments in the main results.
> > >
> > > If we have adequately addressed your concerns, we kindly ask that you consider adjusting your score taking into account our revision.
> > >
> > > Thank you.

---

> > > > ### Comment · Reviewer_VbEW · 2024-11-27
> > > >
> > > > Thank you for your response.
> > > >
> > > > The motivation behind this method is interesting, and the methodology seems sound. However, my limited familiarity with prior methods in this field and the broader impact of this approach on the community makes it difficult to fully evaluate its significance. While I lean toward recommending acceptance, I defer the final decision to the other reviewers and area chairs.

---

### Official Review · Reviewer_yrhN · 2024-10-28

**Soundness:** 2
**Presentation:** 3
**Contribution:** 2
**Rating:** 3
**Confidence:** 4

**Summary:**

This paper proposes Articulate-Anything, a closed-loop vision-language actor-critic system that is capable of converting real objects (provided via image, video, or text descriptions) into Python codes and further compiles them into URDFs. The proposed approach is evaluted against baselines of previous proposes and noticable improvements over success rates of link placement and joint prediction are achieved.

**Strengths:**

1. The paper is well-motivated. Understanding the structure of articulated objects is important for vision and robotics tasks.
2. The idea of closed-loop iterative refinement is interesting and technically sound.
3. The approach is able to handle diverse input modalities, including texts, images, and videos.
4. It is a good idea to use structural representations like code to model articulation structures.

**Weaknesses:**

1. It is not explained how the retrived part meshes do not conflict with each other. For example, in Fig.4(a), the code fragments do not show mechanisms preventing the 'door' having collisions with the drawer in the 'furniture_body'. In other words, how is the detailed position of the 'door'  (including its height) determined?

2. Some technical details are not clear. For example, the target affordance extraction part in Sec.4.4 and how link placement / joint prediction works for text / visual inputs, as the mesh retrieval results of text / visual inputs are different. A pseudo code can be help.

3. It seems that the method and experiments do not support well the task studied in this paper. According to Sec.3, the task is consistent with a formulation of reconstruction task. However, the proposed approach reterives similar meshes from a dataset rather than reconstruct the mesh in Sec.4, which cannot ensure the consistency between the retrieved mesh and the real object part and may result in large L_mesh. And the experiments just quantitatively evalutate the accuracy of link placement and joint prediction. Evaluations with important metrics like CD for reconstruction are not provided.

4. The robotic application is not solid enough. The test results like the success rate of different manipulation tasks are not provided, making it difficult to give a proper evalulation on the effectiveness of the generated assets. And additional real-world experiments will make this point stronger.

**Questions:**

1. According to Fig.3, a template object is retrieved when visual input is provided. How do the follow-up link prediction and joint prediction steps work on template objects? I assume that the link and joint information of the template object are already provided in PartNet-Mobility dataset.

2. What are the 'dimensions' given by the Layout Planner based on? How to ensure that the dimensions of multiple object parts given by the Planner are not contradictory?

3. Using VLMs to directly output numerical values (dimensions, angles, etc.) seems not a good idea for embodied applications which have high demands on safety and reliability, as studies have shown that LLMs are not strong enough to process numerical operations.

---

> ### Author Response · Authors · 2024-11-23
> **Authors' Response (1/2)**
>
> Thank you for your thorough and constructive feedback! Here we addressed your comments:
>
> __How to prevent collision between part meshes?__
>
>  Our API performs placement by aligning the child and parent's link centers along an axis and performing collision checks to ensure tight placement without intersection.
> Collision check was mentioned in line 196. We have added a more detailed explanation in line 247 to make it clearer.
>
> __More clarification on affordance extraction__
>
> Our system intelligently identifies which parts need articulation based on demonstrated interactions (Sec. 4.4). For example, given a video showing a bottom pane of a double-hung window being closed (Fig. 2, pane 3.5): (1) the VLM
> analyzes video to understand which part is being manipulated, (2) It then identifies the corresponding part in the 3D model's segmentation map, and (3) Only the relevant part (bottom pane in blue) is selected for articulation, rather than annotating all movable parts.
>
> __More clarification on the mesh retrieval process__
>
> The mesh results for text and visual inputs have the same abstraction. As Fig. 2 (pane 2) shows, the object is broken into individual part meshes. Even when a template object is retrieved
> in the case of visual inputs, the object is still broken into parts to ensure a unified API.
>
> __Why was retrieval used instead of mesh generation and can reconstruction method be computed?__
>
> Thank you for your suggestion! Although the initial results were obtained via retrieval, as we stated in the conclusion: our primary technical contribution lies
> in the articulation process, and our components can be integrated with any mesh acquisation methods. We have additionally included new results replacing the retrieval mechanism with meshes generated by a diffusion model. Please see our website and paper Appendix A.6. for the new results.
> We also quantify our superior performance using Chamfer distance in Table 2 (appendix A.5). Articulate-anything with mesh generation is far superior than all baselines. Even with retrieval, our mesh reconstruction quality is still better than prior works.
>
> We'd also like to note that retrieval alone is sufficient for most applications. We have massive datasets of static 3D models (e.g., Objaverse with +10 million objects) but only minuscule datasets of articulated objects. Simply being able to tap into these static datasets and articulate them automatically would already unlock many potentials in robotics and other applications.
>
> __The robotic application is not solid enough. There is no real-world robotic results.__
>
> We have spent significant effort developing and setting up the control stack on our hardware robotic platform. Our experiment shows that articulate-anything can accurately generate simulated assets from real-world videos, which can then be used to train an RL agent in simulation. The trained policy can be deployed successfully on a real Franka arm, completing all tasks. In comparison, a manual annotation by domain experts using [RialTo](https://arxiv.org/abs/2403.03949) took humans an average of 40.58 minutes per task to complete. Please see our [website](https://iclr-2025-4168.github.io/) for the video demonstration and refer to the updated Sec. 6 in the paper.

---

> > ### Author Response · Authors · 2024-11-23
> > **Authors' Response (2/2)**
> >
> > __Don't template objects already have annotated information? How does link and joint prediction work on them?__
> >
> > Our method is a general-purpose automatic articulation system, which can annotate static meshes from existing massive datasets or completely new generated meshes, as shown in the new results in A.6. The reviewer is correct that the PartNet-Mobility dataset comes with human annotations. As we stated in the experiments, the annotations are used only as ground truth to evaluate various methods. Those annotations are masked out before being fed into our model.
> >
> >
> >
> > __What is the `dimension` output and how to ensure consistency?__
> >
> > The `dimension` is the width, length, and height of an object. The reviewer is correct that consistency can be an issue in general. For example, our method might generate a big drawer that would not fit into the cabinet body. However, we have a comprehensive system prompt that includes many guidelines and examples to prevent this.
> > For example, below is a snippet of the instruction:
> >
> > ```
> > ...
> > 3. Give a realistic estimate of the final dimension of the object in the response. When assembled, your parts MUST stay within the final dimension estimate.
> > ...
> > 6.
> > - Think CAREFULLY about how your parts contribute to the final dimensions of the object. Give calculation for each dimension.
> > - For example: If you intend on creating an object of size [1,0.25,2] from stacking two objects over the y-axis, you **SHOULD and MUST create two parts of dimensions [1,0.125,2] and [1,0.125,2] INSTEAD OF [1,0.25,2] and [1,0.25,2]**. Doing the latter is INCORRECT and would result in double the dimension along the y-axis.
> > ...
> > 7.
> > - Follow the examples to give justification for the dimensions of the parts, especially how they contribute to the final dimensions of the object body.
> > - E.g., vertically stacked parts should generally add up to the height of the object body/frame ect.
> > ...
> > ```
> >
> > __Safety and reliability with VLM generated outputs__
> >
> >  We appreciate this important concern about numerical precision in embodied applications. Our system addresses this through a multi-layered approach:
> > 1. First, As the reviewer pointed out, when evaluating our strengths, we use "structural representations like code" to articulate objects. We leverage VLMs' strength in high-level planning while delegating precise operations such as collision checks and kinematics to a classical pipelines,
> > avoiding pitfalls in prior works such as URDFormer's direct 3D coordinate prediction.
> > 2. Second, we introduce a critic component to evaluate and correct our outputs, adding another layer of verification.
> > 3. Lastly, our work generates 3D assets in simulation. We are not, for example, generating the torque action for a real-world robot directly. Deploying simulation policy to the real-world in general involves many intermediate safety mechanisms. For example, we implement torque limits and trajectory smoothing on our real-world Franka arm.
> >
> > As with many prior works ([DrEureka](https://arxiv.org/abs/2406.01967), [OpenVLA](https://arxiv.org/abs/2406.09246), ect) that integrate VLMs with a robotic pipeline, we have also deployed
> > our trained policies safely on a real Franka arm (Sec. 6).

---

> > > ### Author Response · Authors · 2024-11-25
> > >
> > > Dear Reviewer yrhN,
> > >
> > > Thank you again for your review. As we near the end of the discussion period, we kindly ask that you review our response to your feedback and consider adjusting your score taking into account our revision.
> > >
> > > Thank you.

---

> > > > ### Comment · Reviewer_yrhN · 2024-11-26
> > > >
> > > > Dear Authors,
> > > >
> > > > Thank you very much for the response. I noticed that in the revised version, there are many more supplementary materials are provided in the appendix and website.
> > > >
> > > > However, some of the concerns raised in my official review have not been addressed.
> > > >
> > > > >W1. How are the detailed numerical parameters (*e.g.*, 6-dof pose, size) of a part determined?
> > > >
> > > > Thank you for the explanation about collision check. I agree that using collision check can remove the results with wrong parameters produced by the link placement and joint prediction module, however, collision check cannot provide the results with correct parameters intuitively. A discussion on how the correct parameters are obtained would be helpful.
> > > >
> > > > >W2. When a template object is retrieved in the case of visual inputs, the object is still broken into parts.
> > > >
> > > > Why the object retrieved in the case of visual inputs is broken into parts first? Since the proposed approach finally need to use link placement and joint prediction to combine the parts of the retrieved object to an object, this process seems unnecessary and confusing.
> > > >
> > > > >W3. Evaluation with CD.
> > > >
> > > > Thank you for the results in Tab.2, but why the experiment setting with CD as metric is different from the setting of link placement and joint prediction experiment?
> > > >
> > > > >W4. Object Manipulation Experiment in Robotic Application.
> > > >
> > > > Thank you for the video demonstrations. However, for a solid robot manipulation experiment, I suggest that there are many important details are still missing in the revised version. The authors should carefully check the details that are crucial for other researchers to assess and reproduce the results and conduct subsequent studies, and provide them with clear statement in the paper.
> > > >
> > > > Some of these details are listed below:
> > > >
> > > >     What are the definitions of manipulation tasks? How many objects per category are used in the experiments? How many trials are conducted on the same object?
> > > >
> > > > >Q1. Link and joint annotations are masked out before being fed into our model.
> > > >
> > > > Thank you for the response. I am still confused that since the retrieval result (selected candidate) for visual input is an object as shown in Fig.3-a, how to mask such annotations on the selected candidate?

---

> ### Author Response · Authors · 2024-11-26
>
> Dear Reviewer yrhN,
>
> Thank you for your detailed response! We provide more detailed clarifications below. Please let us know if anything is still unclear and we’ll be happy to follow up.
>
> __W1. How are the detailed numerical parameters (e.g., 6-dof pose, size) of a part determined?__
>
> We will add this discussion along with relevant Python code to the Appendix.
>
> __Part positioning:__ Our system determines the positions between a child and a parent part by aligning the child and parent centers along a specific axis using their bounding boxes. For example, when placing a child "below" a parent:
>
> ```
> # placing the child center directly below the parent's center along the z-axis
> # x, y same as parent
> target_position[2] += parent_aabb_min[2] - \
>                 child_aabb_max[2] - clearance
> ```
> Collision check is applied afterwards.
>
> Similar to prior works like URDFormer, we do not predict orientations. This design choice is motivated by a key insight: when all objects in the dataset are preprocessed to face the same canonical direction, the parts that should be combined together are already properly oriented. The real question is where to put them. For example, should the the suitcase handle be going on top or below the body?
>
>
> __Part sizing:__ For visual inputs, dimensions come directly from retrieved templates. For text inputs, our Layout Planner uses comprehensive prompting (detailed in our previous response) to ensure dimensional consistency.
>
> __W2. When a template object is retrieved in the case of visual inputs, the object is still broken into parts. This process feels unnecessary.__
>
> In articulated object modeling, breaking objects into parts is necessary for annotating fine-grained movements between parts. Joint prediction gives movement to otherwise static parts and is always required. The question then is: why must we also predict the parts' locations (link placement)? There are two primary reasons:
>
> __API consistency:__  For text inputs where we retrieve individual parts, link placement is clearly necessary. By applying the same part breakdown process to template objects, we ensure a streamlined and unified API across all input modalities - a point raised in the reviewer's earlier questions.
>
> __Context for detailed articulation:__  Consider the left and right doors in Fig.4b. These doors open differently: the left door opens outward from right to left, while the right door from left to right.  For accurate joint prediction, we first ask the system to perform link placement (Fig. 2, pane 3, and Fig.4a). Then, we provide the link placement code for the VLM to fill in the joint code. This creates crucial context for more accurate articulation (e.g., if the link placement places the "door_1" to the left then joint actor knows that "door_1" variable explicitly refers to the left door in the Python code).
>
>
> __W3. Evaluation with CD.__
>
> The main link and joint prediction experiment is done on the PartNet-Mobility dataset while CD is computed on real-world objects. Since our system and some prior works (e.g., urdformer) retrieve objects while others (e.g., real2code) do not, it might be hard to ensure a fair evaluation for all baselines in PartNet. The true test for all methods is whether we can accurately articulate real-world objects where no gt meshes are available to any method so CD comparison on real-world objects is definitely a fair game for all methods.
>
> We'd also note that CD is still a crude metric that does not take into account texture. We thus provide qualitative examples in Fig. 7 and the website. It's evident that our retrieved and generated assets are of markedly higher quality that the baselines.
>
>
> __W4. Object Manipulation Details in Robotic Application.__
>
> Thank you very much for the suggestion! We'll add these details to the Appendix:
>
> - Task definitions and reward functions
> - Per-object trial counts (5 trials per object) with different initial object poses
> - One object per type
> - Training details (3 random seeds, 2M steps each)
> - Learning algorithm (PPO) and hyperparameters
> - Policy architectures and obs/action spaces
> - Hardware details
>
> We’d appreciate it if you can let us know if there is any details we might have overlooked.
>
>
> __Q1. Link and joint annotations are masked out before being fed into our model. How to mask such annotations on the selected candidate?__
>
> To clarify: "masking annotations" means we remove all link and joint information from the retrieved URDF files, keeping only the geometric mesh data. This is in response to the reviewer's observation that PartNet-Mobility objects already contain human-annotated link and joint information. Our system then predicts spatial arrangements and joint parameters from scratch, without access to any human-annotated ground truth.
>
>
> Thank you for your constructive review and continued engagement! We are happy to address any further questions or concerns.

---

> > ### Comment · Reviewer_yrhN · 2024-12-01
> >
> > Dear Authors,
> >
> > Thank you very much for the answers. And I have some more questions.
> >
> > >W1
> >
> > 1. Part position
> >  - It seems that the position of the part is calculated by rules. How many possible circumstances are there of part positioning (*e.g.*, below)? What kind of rules need to be implemented for different circumstances?
> >  - For parts with irregular meshes, their shape cannot be well represented by a bounding box. Will this pose a problem when part positioning? What is the solution?
> >  - If the collision check turns out that the part's position will cause a collision, how the position is corrected?
> >
> > 2. Part orientation
> >  - Thank you for providing this insight. I agree that it is reasonable in most cases, but there are still situations where this does not hold true (e.g., some handles are inclined). It would be great if this issue could be addressed systematically.
> >
> > >W2 & Q1
> >
> > Thanks for the detailed answers. However, I am still confused that as the retrieval result only contains the geometric mesh data, how to break the whole mesh into parts without ground-truth annotations?
> >
> > >W3
> >
> > By only using objects in the training set of PartNet-Mobility for retrieval, the evaluation can be fair for different baselines. I hope the authors could explain why they consider it might be hard to ensure a fair evaluation for all baselines in PartNet-Mobility.
> >
> > Compared to real-world objects, which are not easily accessible to other researchers, PartNet-Mobility objects are publicly available. Therefore, the lack of evaluation on PartNet-Mobility could undermine the reproducibility of this work.
> >
> > >W4
> >
> > I suggest that responding in the form of a bullet list is still insufficient for detailing the experimental setup.
> >
> > If the authors could present these details in more depth in the response, I would be happy to offer suggestions to help improve this aspect.

---

> ### Author Response · Authors · 2024-12-01
> **Authors' response (1/2)**
>
> Dear Reviewer yrhN,
>
> Thank you very much for your insightful questions! We address your questions below and are happy to follow up if anything is unclear.
>
> __W1__
>
> __Part positioning__
>
> - Our VLM is only responsible for high-level planning while the API implements the precise positioning. We implement `above, below, left, right, front, back, inside` for the placement in similar fashion as the code snippet for `below` provided above. Note that without loss of generality, this allows a part to be positioned anywhere in the 3D space because (1) the API takes an optional `clearance` (offset) and (2) the python `place_relative_to` calls can be chained together.
> - Thank you for the insightful observation. Indeed, the major problem with irregular meshes that we have encoutered is "loose" placement. In the left of this [image](https://imgur.com/a/aqBBHfy), you can see that because the faucet's nose is concaving up, placing the spout `above` the body based on bounding boxes alone creates a large air gap. The solution is to perform iterative movement with collision checks to "ensure tight placement without intersection" (Line 248). Intuitively, this means just moving the faucet down the z-axis by a small increment until we detect a collision. Below, is the shortened python code. The iterative collision movement and check is the `snap_to_place` function. With this function, we ensure tight placement and small link error as shown on the right of the image.
>
> ```
> def place_relative_to(self, child_link_name, parent_link_name,
> 	placement, clearance=0.0, snap_to_place=True):
> 	...
> 	if placement == "above":
> 		target_position[2] += parent_aabb_max[2] - \
>                 child_aabb_min[2] + clearance
>             axis = 2
>             direction = -1
>     ...
>     self.move_child_to_target_position(target_position)
>     if snap_to_place and placement != "inside":
>     	self.snap_to_place(...)
>
> def snap_to_place(self, ..., max_iter=100, increment=0.01):
> 	collision = self.is_collision(child, parent)
> 	while not collision and i < max_iter:
> 		target_position = np.array([0.0, 0.0, 0.0])
> 		target_position[axis] += direction * increment
> 		... # move child to target position
> 		collision = self.is_collision(child, parent)
> ```
> - `snap_to_place` stops as soon as a collision is detected. We have not encountered any case where a placement encounters collision right off the bat.
>
> We will add these details and Python code to the Appendix in the camera-ready version.
>
> __Part orientation__
>
> We note that the part meshes in the dataset are extracted from some parent objects (i.e., a handle part always belongs to _some_ suitcase object). Thus, if the objects are preprocessed to face some canonical direction, orientation between parts should not be an issue. In general, for a systematic treatment, a rotation degree argument can also be added to `place_relative_to` to control orientation while prompting the VLM accordingly but we have found this to be unnecessary.
>
> __W2 & Q1: Part Segmentation__
>
> We do not use link and joint annotations (no locations, no movement) but we do use ground-truth segmentations provided in the PartNet-Mobility dataset to break whole objects into parts. The availability of part segmentation is assumed in many retrieval-based works (e.g., [Singapo](https://arxiv.org/abs/2410.16499), [URDFormer](https://arxiv.org/abs/2405.11656), [CAGE](https://arxiv.org/abs/2312.09570), [NAP](https://arxiv.org/pdf/2305.16315), ect)
>
> As we responded to Reviewer bBwZ, in future works when applying our method to a larger dataset like Objaverse where part segmentation might be sparsely available, or to newly generated meshes from a diffusion model (Fig. 7, 22), an additional segmentation step is required.  Particularly, in our supplemental mesh generation experiment, we first render multi-view RGBD images of the 3D model in simulation, and run SAM to obtain 2D segmentation masks. The 2D masks are lifted to 3D via projective geometry using camera parameters and depth maps. Once the 3D part segmentation is complete, we run our articulation method as described in the paper.
>
> We will include these details to make it clearer in the camera-ready version.
>
>
> __W3 Chamfer distance on PartNet-Mobility evaluation__
>
> Thank you for this valuable suggestion! Our method requires no training instead relying on a few in-context examples (Fig. 10). So to make the comparison fairer, when evaluating each object, we remove that object from the candidate pool, preventing our system to retrieve the _exact_ object. The result on PartNet-mobility, to be added to the paper, is included below (mean and std). The violin plot is included [here](https://imgur.com/a/wng4cUi).
>
> | Method | Chamfer distance |
> |--------|-----------------|
> | Articulate-Anything (retrieval) | 0.1007 +/- 0.062 |
> | Real2Code (Oracle) | 0.229 +/- 0.166 |
> | URDFormer (Oracle) | 0.429 +/- 0.267|
> | URDFormer (DINO) | 0.437  +/- 0.217 |

---

> ### Author Response · Authors · 2024-12-01
>
> __W4 Robotic Details__
>
> As we can no longer update the paper with a revision, we will provide the text below. As before, please let us know if there is any details we might have overlooked.
>
> __Task definitions__
>
> We experiment with 4 tasks: closing a toilet lid, closing a laptop, closing a microwave, and closing a cabinet drawer. We first train a RL policy in simulation using Robosuite and Mujoco \citep{zhu2020robosuite, todorov2012mujoco}. The robot is a Franka Emika arm with 7 degrees of freedom and a gripper. The dense reward functions are manually designed to: (1) move the arm closer to the object (reaching reward), (2) reduce the object's target joint state (angle reward. e.g., making the toilet lid closed), (3) prioritize using the gripper and (4) penalize for sudden movements. The reward is provided below.
>
> ```
> def reward(self, action=None):
> 	reward = 0.0
> 	if self._check_success():
>         reward = 100
> 	elif self.reward_shaping:
>         # reaching reward
>         obj_pos = self.sim.data.body_xpos[self.obj_body_id]
>         gripper_site_pos = self.sim.data.site_xpos[self.robots[0].eef_site_id]
>         cur_dist = np.linalg.norm(gripper_site_pos - obj_pos)
>         if cur_dist < self.closest_dist:
>             self.closest_dist = cur_dist
>             reaching_reward = 1
>             if cur_dist < 0.1:
>                 reaching_reward = reaching_reward * 0.25
>             reward += reaching_reward
>
>         # angle reward
>         if self.first_frame:
>             self.first_frame = False
>             self.prev_angle = self.sim.data.get_joint_qpos(self.targetted_joint)
>         else:
>             current_angle = self.sim.data.get_joint_qpos(self.targetted_joint)
>             diff_angle = current_angle - self.prev_angle
>             angle_reward = 0
>             if np.abs(diff_angle) > 0.001 and self.sim.data.time > 3:
>                 if diff_angle < 0:  # Reward for closing
>                     angle_reward = 2.25
>                 elif diff_angle > 0:  # Penalty for opening
>                     angle_reward = -1
>             reward += angle_reward
>             self.prev_angle = current_angle
>
>         # penalize if any joint is closer to the object than the gripper
>         for joint_name in self.robot_joint_names:
>             joint_pos = self.sim.data.get_joint_qpos(joint_name)
>             joint_obj_dist = np.linalg.norm(joint_pos - obj_pos)
>             if joint_obj_dist < cur_dist:
>                 joint_pose_penalty = -5
>                 reward += joint_pose_penalty
>                 break
>
>         # penalize for large joint velocities
>         joint_velocities = [self.sim.data.get_joint_qvel(
>             joint_name) for joint_name in self.robot_joint_names]
>         joint_velocities = np.array(joint_velocities)
>         joint_velocities = np.abs(joint_velocities)
>         if any(joint_velocities > 2) and self.sim.data.time > 0.5:
>             action_penalty = -0.5
>             reward += action_penalty
>
>     return reward
> ```
>
> __Policy Learning:__ In simulation, we follow a teacher-student distillation training (\citep{eurekaverse, chen2020learning}). We first train a privileged state-based teacher using proprioception and object's pose and targeted joint's state. The policy is trained using the reward above with PPO \citep{schulman2017proximal}. We then distill a student policy that takes in depth frames from a side-view camera and proprioception via behavioral cloning. Visual inputs are encoded via a 2-layer convolutional neural network while non-visual inputs such as proprioception are encoded via a 2-layer feedforward network. The features are concatenated and fed to a 2-layer feed-forward network to produce actions.
>
> __Domain randomization:__ The physical parameters such as friction, stiffness, mass and damping are randomized. The camera viewpoints are also slightly deviated. The object's pose and scale are randomized. The initial target joint state is randomized (e.g., how far the drawer is already opened). The depth observations are augmented by adding Gaussian noise with random pixel blackouts (setting some pixels to 0). During distillation, we randomly delay the action and depth sensing by 60-100ms and the depth frames are updated at 30Hz.
>
> __Sim2Real Transfer:__ The policy outputs the position of each joint and a binary gripper command for closing and opening. The delta arm joints are limited to (-0.5, 0.5) for safety. The student policy is deployed zero-shot to a Franka Emika Robot with depth frames captured by a side Zed camera. The depth frames are center-cropped to reduce background noises. We use one object per task. Each task is rolled out 5 times with varied initial object pose and target joint state.

---

> > ### Comment · Reviewer_yrhN · 2024-12-03
> >
> > Dear Authors,
> >
> > Thank you for the additional answers. I have a few follow-up questions.
> >
> > >W1 - Part Positioning - Irregular Meshes
> >
> > How do the authors calculate the position on the x- and y-axes (target_position[0] and target_position[1])?
> >
> > I suggest that, in the faucet example shown in the image (for clarity, I denote the top part as **Part A** and the bottom part as **Part B**), the x, y coordinates of the center of the cylindrical pillar of both Part A and Part B should be aligned. This issue does not seem to be addressed in the authors' response.
> >
> > >W4
> >
> > I suggest that an important point has been overlooked, namely, what the criteria are for determining the success or failure of an operation for different tasks.

---

> ### Author Response · Authors · 2024-12-03
>
> Dear Reviewer yrhN,
>
>
> __W1__: The x and y-axes are the same as the parent link when the placement is `above` and `below` (i.e. vertical movement). Similarly, for `front` and `back`, only the y-axis is changed (x, z are aligned). And for `left` and `right`, only the x-axis is changed (y, z are aligned). In the example, as the reviewer correctly pointed out, the x, y coordinates of part A and part B are indeed aligned (Part A is a child of part B in this case). The `target_position` is first set to the parent's center. Then, based on the placement, only one coordinate is changed. For brevity, we only provided a _shortened_ code snippet in our earlier response. But we will add the full code to the appendix in the camera-ready version for clarity.
>
> __W4__: Thank you for bringing this up! Take the task of closing a toilet lid for example. In simulation, the toilet lid is represented via a revolute joint ranging from 0 (fully horizontal) to 90 degree (fully vertically upright). The check success function involves checking if the joint state is small enough. For tasks like toilet, microwave, and laptop, the joint states are the angle of the lids and for the cabinet task, the joint state is the displacement of the drawer (how far is the drawer opened). For example,
> ```
>     def _check_success(self):
>         """
>         Check if toilet lid has been closed.
>
>         Returns:
>             bool: True if toilet lid has been closed
>         """
>         toiletlid_angle = self.sim.data.get_joint_qpos("toilet_joint_1")
>         return toiletlid_angle < 0.01
> ```
>
> This allows automatic success checking in simulation for training policies. When deploying to the real-world, we monitor and record the trajectory, and determine success manually.

---

> > ### Author Response · Authors · 2024-12-03
> >
> > Dear Reviewer yrhN,
> >
> > As the rebuttal period draws to a close, we'd like to thank you for your thorough, constructive review and valuable suggestions, which have helped strengthening the paper.
> >
> > I have clarified our part positioning, segmentation, and robotic setup in more details, and provided the chamfer distance comparison on the PartNet-Mobility dataset and real-world objects. We will include the detailed discussion of the API and robot setup, and the mesh reconstruction results in the camera-ready version.
> >
> > If we have adequately addressed your concerns, we kindly ask that you consider adjusting your score.
> >
> >
> > Thank you.
> >
> > Authors

---

### Official Review · Reviewer_bBwZ · 2024-10-29

**Soundness:** 3
**Presentation:** 3
**Contribution:** 2
**Rating:** 6
**Confidence:** 3

**Summary:**

*Articulate-Anything* is a system that automates the articulation of complex objects from various input types, including text, images, and videos. It utilizes vision-language models (VLMs) to generate code that creates interactable digital twins compatible with 3D simulators. The system incorporates a mesh retrieval mechanism to leverage existing 3D assets and employs an actor-critic approach to iteratively propose, assess, and refine articulations, self-correcting to enhance accuracy.

**Strengths:**

1. Compared to previous works, this approach can begin with various inputs, including text, images, and videos.

2. By using iterative refinement with an actor-critic model, this method generates articulated assets with greater accuracy than prior approaches.

3. It utilizes vision-language models (VLMs) to perform the task without requiring network training or fine-tuning, eliminating the need for large-scale datasets, which are challenging to obtain for articulated objects.

**Weaknesses:**

1. While VLMs hold the potential to go beyond existing dataset, this method retrieves parts from those datasets. Thus, it is still constrained by existing datasets.

2. The framework is mainly prompting VLMs to construct articulated objects, with limited technical contributions. And the iterative refinement phrase only raise limited accuracy.

**Questions:**

1. What types of inputs does this work use when comparing to prior methods? Are these inputs—such as text, images, or videos—consistent with those used in previous works?

2. In terms of object retrieval, since this work aims to create digital twins from real-world observations, does the generated articulated object accurately resemble the real-world counterpart? Given that retrieval relies on CLIP scores based on text descriptions, the generated objects might be different from the original objects.

3. Could accuracy be reported across different object categories? Performance might vary significantly by category; for instance, simpler objects like windows and storage furniture may be easier to generate, whereas complex items like toilet seats or shopping carts could present more challenges.

---

> ### Author Response · Authors · 2024-11-22
>
> Thank you for your constructive feedback! Here we addressed your comments:
>
> **Retrieval**
>
> Our primary technical contribution lies in the articulation process. As mentioned in the conclusion, articulate-anything is general-purpose and can be integrated with any mesh acquisition method, be retrieval or generation.  We have additionally included new results replacing the retrieval mechanism with meshes generated by a diffusion model. Please see our [website](https://iclr-2025-4168.github.io/) and paper Appendix A.6. for the new results.
>
> We'd also like to note that retrieval alone is sufficient for most applications. We have massive datasets of static 3D models (e.g., Objaverse with +10 million objects) but only minuscule datasets of articulated objects. Simply being able to tap into these static datasets and articulate them automatically would already unlock
> many potentials in robotics and other applications. Even with retrieval, our mesh reconstruction quality is still superior than prior works as shown in Table 2.
>
> **Technical contributions**
>
> Our work introduced three key innovations: (1) the ability to consume more diverse and grounded types of input, (2) modeling articulation as program synthesis, and (3) developing a closed-loop self-improving system. These advances significantly outperform prior work, achieving a sixfold increase in success rates (Fig. 5) with notably better qualitative results (Fig. 7). Our technical components provide substantial improvements over a standard LLM prompting baseline (real2code).
>
> Our iterative refinement makes two important contributions: reliable failure detection (our critic highly correlates with ground truth, Fig. 15) - crucial for robotics applications - and a 2.4% improvement in joint success rate, which exceeds 25% of URDFormer's total performance. In safety-critical applications like robotics, simply being able to detect failure is important. Additionally, we currently use one camera pose for all tasks. Further camera optimizations (e.g., multi-view, zooming into small parts) can improve the actor's success even further
>
>
> **Other questions**
> - Type of inputs:
> >  Our experiments use video inputs when comparing with other methods, as specified in Fig. 7's caption and the experimental details (line 790). While existing methods are designed for specific inputs (URDFormer uses cropped images, Real2code uses text bounding boxes), our approach uniquely supports multiple input modalities.
>
>
> - Does CLIP scores based on text description mean that retrieved assets are not realistic?
> > As detailed in section 4.1 and Fig. 3, for visual inputs, we visually match real-world objects with simulated assets using a VLM. CLIP is used only for text inputs or initial object category filtering. For instance, in Fig. 3,  the VLM selects the 3D model that best resembles the real-world monitor from available monitor assets.
> - Can accuracy be reported across different categories
> > Thank you for your suggestion! We have added the accuracy breakdown across different categories in Fig. 20 and 21 in the Appendix. We use one camera pose for all tasks. As such, we have observed that smaller object parts are generally harder to place correctly, and objects with more esoteric movement are harder to predict joint, as the reviewer alluded to.

---

> > ### Comment · Reviewer_bBwZ · 2024-11-22
> >
> > Thanks for the detailed response. There are enough technical contributions for this work.
> >
> > However, I have some more questions.
> >
> > First, when comparing with other method, you are using videos as input while others are using text or images. I believe this is not a fair comparison since video is much harder to acquire comparing to text or images. It would be better to compare your scores with purely text or image input separately. Or augment with text to video model to make it a whole pipeline (it might not work...).
> >
> > Secondly, for the mesh retrieval, it doesn't convince me to select from objaverse. There's way more objects in objaverse and is not categorized, unlike parent-mobility. Also, I think we are retrieving parts rather than a whole object, which does not exist a lot in objaverse.
> >
> > Finally, for the mesh generation part, how is the pipeline actually working? It is said in the appendix that you generate a static mesh and then articulate it? How to articulate it using your existing pipeline?

---

> ### Author Response · Authors · 2024-11-23
>
> Thank you for your thoughtful questions! We address each of the questions below.
>
> __Different modality comparison__
>
> Our experimental results in Fig. 6 demonstrate strong performance across input modalities. Even with just text or image inputs, our success rates are still significantly higher than prior works. While this ablation is only run on the largest object category due to budget constraints, these results are further reinforced by the real-world results in Fig. 7. For real-world objects, our text and image-based methods succeed on some tasks whereas the baselines consistently fail.
>
>
> Regarding the ease of data acquisation, our system handles casual, unconstrained inputs whereas prior works require heavy data curation as detailed in Appendix A.4. For example, as shown on our [website](https://iclr-2025-4168.github.io/), input videos are captured in cluttered environment (e.g., see laptop), with angled (e.g., window) and titled (e.g., microwave) views with no additional data cleaning. This unlocks the exciting potential of applying the system on internet-scale video datasets such as Ego4D or Youtube. In contrast, URDFormer required manual bounding box correction (Fig. 18), and Real2Code required extensive scene setting, scanning via a Lidar-equipped phone, tuning segmentation masks and hyper-parameter (Fig. 17).
>
> __Mesh retrieval on Objaverse__
>
> Object labels are not a requirement of our approach -- we used them solely as an optional filtering step to reduce computational costs whenever possible. Our system can leverage standard efficient large-scale visual retrieval algorithms (e.g., approximate nearest neighbor, hierarchical embedding, locality-sensitive hashing) that enable search engines to search through billions of images. Secondly, having part meshes is also not a requirement. Our method naturally handles both whole objects and their constituent parts, as demonstrated in our mesh generation results, explained in more details below.
>
> __Mesh generation details__
>
> After obtaining a static mesh, we perform a standard 3D segmentation technique to obtain part meshes. Particularly, we first render multi-view images of the 3D model in simulation, and run SAM to obtain 2D segmentation masks. The 2D masks are lifted to 3D via projective geometry using camera parameters and depth maps. Once the part meshes are obtained, we apply our method as described in the paper.
>
> Please let us know if you have any more questions that we can address!

---

> ### Comment · Reviewer_bBwZ · 2024-11-23
>
> "Even with just text or image inputs, our success rates are still significantly higher than prior works."
> Is there any experiments for that? All what I can see is figure 6. Inferring from that, with text input, Link Placement has 68.6% successful rate and Joint Prediction has 48.9% successful rate. Those two time up to 34% success rate, which is only 10% higher than URDFormer in Figure 5? And in Figure 5, they are tested on all seen categories rather than just on largest category.
>
> Can you just provide a single experiment comparing with baselines with fair input modality? (maybe just compare to urdformer to save time)
>
> About "Mesh generation details", I think it's not trivial to lift multi-view 2D masks into 3D. Also, it's not reasonable when you have only the surface mesh (as we can see there's visual artifacts in your website drawer demo video). But I'll stop arguing this part since all previous works are also generated by retrieving object parts, and this doesn't hurt the major contributions of this work.

---

> ### Author Response · Authors · 2024-11-24
>
> Dear reviewer bBwZ,
>
> We've included an additional experiment in Appendix A.7 (Fig. 23) as requested. Please see the revised draft. To save time, this ablation was run on the corresponding seen classes of Real2Code and URDFormer. Articulate-Anything's text and image-based, while degrading compared to our main video-based method, still significantly outperforms the comparable baselines.
>
> We'd also like to note that "Those two time up to..." should not have been done. The joint prediction success rate **already** factors in the link failure (Line 308). As you can see from the left-most pane of Fig. 8, joint prediction failures already include link failures.

---

> > ### Author Response · Authors · 2024-11-25
> >
> > Dear Reviewer bBwZ,
> >
> > Thank you again for your valuable feedback. We have included mesh generation and an comparison with the same modalities in the paper.
> >
> > If we have adequately addressed your concerns, we kindly ask that you consider adjusting your score taking into account our revision.
> >
> > Thank you.

---

> ### Comment · Reviewer_bBwZ · 2024-11-27
>
> Dear Author,
>
> The unfair comparison part has been well addressed. Hopefully the author could state that clearly in the camera ready version.
>
> One of my other concern is about the technical contribution. The major component of this paper seems to be the iterative refinement part. While it is 2.4% growth in the success rate, it seems to be subtle comparing to your ablation score (over 70%).
>
> Also, the success thresholds for link placement and joint prediction are also quite weird: 0.25 rad is large enough to make a difference. Is that one of the limitation of this work since it's hard for VLMs to sense smaller changes?
>
> In all, I personally still don't think prompting VLMs to place retrieved meshes and assign joints has enough technical contribution. However, that might not be the community standard. Thus, I raise my score to borderline accept and will let other reviewers and AC to judge that.

---

> ### Author Response · Authors · 2024-11-27
>
> Dear Reviewer bBwZ,
>
> We sincerely thank you for your thorough review and constructive suggestions throughout this process.
>
> We will include, if space permits, or at least reference the new modality results in the main text.
>
> Regarding your concern about thresholds, we conducted additional analysis with a much stricter threshold of 0.01 radians (compared to the original 0.25):
>
> | Method | Success Rate (%) |
> |--------|-----------------|
> | Articulate-Anything | 52.1 |
> | Real2Code (Oracle) | 11.9 |
> | URDFormer (Oracle) | 14.6 |
> | URDFormer (DINO) | 8.7 |
>
> Even with this significantly stricter threshold, our method maintains a 3.5-6× improvement over baselines. Most failures at this threshold occur with very fine-grained and small movements (e.g., keyboard buttons). _These errors are typically not discernible to human eyes at our current camera pose._
>
>
> Previously, the reviewer acknowledged that "There are enough technical contributions for this work." So we would like to re-iterate our contributions beyond iterative refinement:
>
> - __State-of-the-art performance:__ Our method achieves sixfold improvement in quantitative metrics (Fig. 5) with substantially better qualitative results (Fig. 7).
>
> - __Multi-Modal Flexibility:__ We uniquely support in-the-wild inputs through text, images, and videos, eliminating the need for specialized data like point clouds or connectivity graphs.
>
> - __Flexible Asset Generation:__ Our approach works with both retrieved and generated meshes (Fig. 7, Table 2), producing high-fidelity digital twins that closely match real-world objects.
>
> - __Scientific Insights:__ Not only have we shown that the method is highly effective, through extensive ablation studies, we have provided scientific insights:
>   + __In-context learning:__ Our training-free approach, as you noted, eliminates "the need for large-scale datasets, which are challenging to obtain for articulated objects." Through a few in-context examples alone, we achieve dramatic performance improvements from 4.6% to 77.7% (Fig. 10).
>   + __Richer input modality:__ Our system leverages richer input modalities to improve accuracy from 48.9% to 77.7% (Fig. 6). This capability enables crucial disambiguation—for example, while a window door could theoretically use either a revolute or prismatic joint, video input allows our system to determine the correct joint type (Fig. 7).
>
>   + __High-level API:__ Rather than directly predicting numerical values like previous approaches, we leverage high-level planning strength of VLM while delegating low-level details like kinematics to a classic pipeline. This architecture demonstrates superior performance against baselines even when using identical input modalities (Fig. 23).
>   + __Self-evaluation and correction:__ We introduce the first self-evaluating articulation system that can detect failures — a critical feature for robotics applications. This capability enables autonomous refinement, yielding significant improvements in both link placement (5.8%) and joint prediction (2.4%). While the joint gain is more modest, it is because joint prediction task is harder. More camera pose optimization is likely to increase this gain.
>
>   Thank you again for your careful review and consideration!

---

> > ### Comment · Reviewer_bBwZ · 2024-11-28
> >
> > Dear Authors,
> >
> > I do appreciate the significant contributions of your work, including the notable performance improvement, the integration of multi-modality input, and other advancements, which is why I rated it as a *weak accept*.
> >
> > In my last response, my concern was not about *what* you achieved but rather *how* you achieved it. While the **state-of-the-art performance**, **multi-modal flexibility**, and **scientific insights** are commendable, they are largely a result of transitioning from "learning a network to generate" to "building a VLM pipeline to generate" without introducing fundamentally new techniques.
> >
> > Additionally, while you highlight "significant improvements in both link placement (5.8%) and joint prediction (2.4%)," I question how significant these improvements are relative to your original scores, which are already above 70%. Could you provide more context or evidence to substantiate the significance of these improvements?
> >
> > Regarding **Flexible Asset Generation**, as I previously mentioned, I do not believe this component should be emphasized in the current paper for the following reasons:
> > 1. At the time of submission, the paper did not mention this generation functionality. Adding such a substantial new pipeline at this stage is beyond the scope of the review process. I would recommend incorporating these components and resubmitting to a future conference.
> > 2. The methodology for generating the inner structure of the objects remains unclear. If you wish to present this as a contribution, the paper should provide a detailed explanation of the generation and segmentation pipeline, along with more qualitative results to support the claims.
> >
> > Finally, I believe the latest response still calculates scores based on video input. As I stated earlier, this does not constitute a fair comparison. Similarly, the claim of a sixfold improvement is overstated, as the input modalities are fundamentally different.

---

> ### Author Response · Authors · 2024-11-28
>
> Dear Reviewer bBwZ,
>
> Regarding the critic component, as we detailed in two previous responses, this component uniquely enables us to detect failures and improves the system. The improvement is in excess of 25% of URDFormer's performance. We believe further camera pose optimization (e.g., multi-view, zooming into small parts) will increase the gains even further.
>
> As we stated in our contributions in the introduction and our rebuttal, our main technical contribution is in the articulation process. We consider mesh acquisition front-end to be secondary as our method can work with either retrieved or generated assets. We will add more details on the generation process to the paper as suggested.
>
> Lowering the threshold was done on the main result. We have included an ablation on comparable modality as suggested in Fig. 23. Nonetheless, we believe that videos is the correct format to describe motion. Conceptually, the window example in Fig. 7 demonstrates that impoverished input modalities fundamentally cannot capture the necessary information for joint articulation in all cases. Further, as argued in our first response, we believe our method to be much more easily and automatically deployable to large-scale datasets as prior works require heavy manual curation.
>
> Once again, we thank you for your constructive review,  perspectives and for raising our score. Your suggestions have helped strengthen the paper, and have been incorporated into the revision.
>
> Best regards,
>
> Authors

---

> > ### Comment · Reviewer_bBwZ · 2024-12-02
> >
> > Dear Author,
> >
> > *The improvement is in excess of 25% of URDFormer's performance:* But it is merely 2.4% improvement in total, which is not significant enough comparing to over 70% successful rate of your work. When doing ablation study, you should compare to the original version of your pipeline rather than previous works.
> >
> > *Our method can work with either retrieved or generated assets:* Your work can only worked with assets that have part segmentation labels, which in my mind is only the partnet dataset.
> >
> > *We will add more details on the generation process to the paper as suggested:* I still can't find it in the paper, is there anything I missed?
> >
> > *Nonetheless, we believe that videos is the correct format to describe motion:*  Sure it contains more information than images and texts. However, it is hard to acquire. Imagine I am collecting a large-scale dataset for articulated objects, I can generate endless images using image generation model and then use the URDFormer pipeline to generate articulated objects. But how can I acquire endless videos? The video generation models are not that powerful yet.

---

> ### Author Response · Authors · 2024-12-03
>
> Dear Reviewer bBwZ,
>
> As the rebuttal period draws to a close, we'd like to thank you for your thoughtful review and valuable suggestions, which have helped strengthening our paper. Below, we address the points that you raised.
>
> Regarding the __performance improvement__: Your observation about comparing against previous works is well-taken. Upon closer inspection, we note that 14.7% of joint failure originates from incorrect link placement in the previous stage (Fig. 8). Thus, the maximum achievable joint accuracy is only 85.3% (not 100%). At the end of iterative refinement, our accuracy is at 75%, about 10% shy of the ceiling. The remaining failure cases are _hard_; they tend to involve small objects with irregular movement like Pliers and Knife (Fig. 21). In these cases, if the actor fails to correctly articulate the object in the first iteration, even the critic mechanism proves insufficient for correction.  We believe that enhanced camera optimization strategies—such as multi-view systems or dynamic zoom capabilities for small objects—could address these limitations, but reserve it for future work.
>
>
> Regarding __segmentation__: as the reviewer previously suggested, most other works (e.g., [Singapo](https://arxiv.org/abs/2410.16499), [URDFormer](https://arxiv.org/abs/2405.11656), [CAGE](https://arxiv.org/abs/2312.09570), [NAP](https://arxiv.org/pdf/2305.16315)) requires part segmentation: which can either be already available in the dataset (e.g., PartNet) or predicted (e.g., Real2code). Our main result is based on retrieval, leveraging existing part segmentations. For our supplemental results with newly generated meshes, we predict segmentations using SAM, similar to Real2code's approach. We explained the 3D segmentation procedure in our earlier response. We will add these implementation details to the camera-ready version.
>
> Regarding __video inputs__: We agree that videos might be harder to acquire. However, we argue that richer input modalities, whenever available, should be used for improved accuracy (Fig. 6). Additionally, our system also supports images and texts, providing flexibility to the user.
>
> In terms of __practicality__, our system offers two key advantages: (1) working with casually captured in-the-wild inputs out-of-the-box, and (2) automatically detect failures (via a critic). These features are important as otherwise, the cost of hiring humans to first extensively curating the data (e.g., manually correcting part bounding boxes, Appendix A.4, Fig. 17-18) and afterwards filtering out incorrect predictions might outweigh the benefits of automatic articulation. Without such automation, manual asset annotation might indeed be more cost-effective.
>
> Thank you,
>
> Authors

---

### Official Review · Reviewer_SpUi · 2024-11-04

**Soundness:** 3
**Presentation:** 3
**Contribution:** 3
**Rating:** 8
**Confidence:** 3

**Summary:**

This paper presents a method for generating simulated 3D objects, with moving parts, that can be inferred with from multi-modal inputs (text, images, videos). This is interesting because is suggests a method of inferring models of objects, so that (robot) agents can be trained on richer and more representative simulations for things that exist in the real world.

The method leverages LLMs to generate URDF descriptions (effectively code) which can then be iteratively refined based on feedback from a critic module. This is a neat use of LLMs that follows a trend in ML, but is well applied here.

Success is primarily evaluated by taking ground-truth URDF descriptions from an existing dataset (PartNet-Mobility), masking out information in them, and then comparing the reconstructed URDF with the ground-truth, using a threshold tolerance to define "success rates".

**Strengths:**

This paper does a lot of interesting things. Most of all, the idea of using LLMs to generate code is well applied here, and carried through evaluations that lead to at least proof of concept robot applications (though I believe still in simulation).

To the best of my knowledge, this is an *original* and *significant* contribution. For example, I don't know the prior work well, but the baselines sound reasonable.

The *quality* and *clarity* of the paper is decent as well, though could be improved (I have some suggestions below).

The multi-modality of the approach is also notable, and I appreciated the ablation of the different modalities (section 5.1).

**Weaknesses:**

Three weaknesses in the paper stood out but could be improved in revision.

The first is relatively minor, but there seemed to be missing references to other works that have used LLM agents in an actor-critic setup.

One paper that came to mind was the Text2Reward paper (https://arxiv.org/abs/2309.11489v3). But more generally, it would help to emphasise that this general idea - which is very cleverly used here - is not part of the paper's novely.

The second weakness is a lack of statistics and confidence intervals in the results.

For example, Table 1 shows average joint prediction errors and it is possible to see differences between these averages. But without statistical testing (e.g. F-test or t-tests) it is hard to appreciate how meaningful these differences are. Some measure of variance would be useful here too (e.g. +/- standard deviation, to appreciate the consistency of the different models). In general, plotting more than summary statistics (e.g. averages) would be much stronger (e.g. violin or raincloud plots).

Relatedly, the Figure 5 caption says that the approach "significantly outperforms all baselines". But as before, there are no statistical tests to back this up - just comparative %success rages without confidence intervals. I agree that the proposed model results appear better, but without a sense of the confidence intervals it's hard to be certain. In this case, Chi-square tests (if the samples are large) or Fisher's exact tests (for smaller sample sizes) might be appropriate.

The final weakness might be harder to address as it concerns fitting all of the main results into the main body of the paper.

To me, it was most notable that the application for robotics section felt incomplete and even confusing. I was not able to properly evaluate what was done or whether it was good based on the short section, which is a shame because it is a very interesting culmination of what the paper builds to. The paper also frames this as a key result, but then doesn't quite get all the way there.

Of course I appreciate that there are page limits, and I see that there are additional details in an appendix. But it would be better to fit more of that into the main paper for completeness.

Similarly, the in the wild results seemed incomplete in the main text as well, though I see there are more details in the appendix.

Unfortunately I don't have any great suggestions on what you could cut to make room for these things. I suppose you might play with resizing some of the figures, cutting words where you can, and reformatting carefully.

**Questions:**

In addition to the questions implied above, could I also ask if you experimented with weighting the losses in Eq 1?

In section 5, on page 6, you also write that "Success requires all criteria to be within a small threshold... The position threshold is set to 50mm and the angular threshold to 0.25 radian (∼ 14.3 degree)."

How did you choose these thresholds? Some defence of or rationale for these choices would be helpful, unless they are arbitrary?

---

> ### Author Response · Authors · 2024-11-22
>
> Thank you for your positive review! We have updated our draft to incorporate your feedback.
>
> **References to other actor-critic VLM/LLM works**
>
> Thank you. We agree that a more detailed discussion of our differences from previous works concerning actor-critic is warranted.
> The [text2reward](https://arxiv.org/pdf/2309.11489v3) paper that the reviewer mentioned uses human feedback while our system utilizes a VLM as a critic. This is a very important distinction as it allows the system to be fully automatic and scales with data without any human effort. Some papers that automate the verification process such as [Code as Reward](https://arxiv.org/pdf/2402.04764) and [Kwon et al 2024](https://arxiv.org/pdf/2310.11604) employ very domain-specific heuristic such as object tracking. There are a few papers that use VLM as a success detector for robotic task such as [Du et al 2023](https://arxiv.org/pdf/2303.07280) and [Guan et al 2024](https://arxiv.org/pdf/2402.04210), providing a binary reward or natural language feedback. However, these are essentially critic-only systems. To our knowledge, we are the first to integrate both VLM actor and critic in a closed-loop system, enabling self-correction over subsequently iterations, all without human efforts.
>
> **Lack of statistical confidences**
>
> We have added standard deviation for continuous errors and 95\% CI for binary errors to Table 1 (Appendix A.5), 95\% CI to Figure 5, and a violin plot in Figure 19 in the Appendix. As before, our superior performance lies far beyond the margin of error.
>
> **Robotics and in-the-wild application**
>
> We have spent significant effort developing and setting up the control stack on our hardware robotic platform. Our experiment shows that articulate-anything can accurately generate simulated assets from real-world videos, which can then be used to train an RL agent in simulation. The trained policy can be deployed successfully on a real Franka arm. In comparison, a manual annotation by human domain experts using [RialTo](https://real-to-sim-to-real.github.io/RialTo/) took an average of 40.58 minutes per task to complete.
> Please see our website for the video demonstration and refer to the updated Sec. 6 in the paper. For in-the-wild results, we have also added the results integrating our system with a mesh generation model to produce more customized assets. Chamfer distance showing superior mesh reconstruction quality is also included in Table 2.
>
> **Other questions**
> - Weighting the loss: as we wrote in Sec. 3, the losses are approximated using a VLM critic. We use few-shot prompting i.e., no explicit training or fine-tuning of the losses.
> - Success criteria: we use a very small threshold as the success criteria. We've found that this threshold is so small that small errors are not noticeable to the human eye. But we also provide the raw errors (unthresholded) in Table 1 where Articulate-Anything is still much better than all baselines.

---

> > ### Author Response · Authors · 2024-11-25
> >
> > Dear Reviewer SpUi,
> >
> > Thank you again for your review. As we near the end of the discussion period, we kindly ask that you review our response to your feedback and consider adjusting your score taking into account our revision.
> >
> > Thank you.

---

> > ### Comment · Reviewer_SpUi · 2024-11-26
> > **Very nice to see qualitative robot results; are there any quantitative results?**
> >
> > Thanks for the work that you have continued to put into this paper. I particularly like the addition of section 6 which links to a video of robots completing the tasks. Do you have any quantitative measures of success that could go into the paper, to show that you didn't run the robot 1k times and then only post the video of the one success?

---

> ### Author Response · Authors · 2024-11-26
>
> Dear Reviewer SpUi,
>
> Thank you for your positive feedback! The RL policies trained in simulation are directly transferred to the real arm where the tasks are fairly simple and deterministic. We trained 3 random seeds per task for 2 million steps and selected the best checkpoints. We run 5 roll-out trials per task with slightly varried object poses. All of these policies are successful in the real-world. Those simulated policies are also included on the website for cross comparison with the real-world execution.
>
> Besides the quantitative measure of the human annotation cost in Fig. 12, we are also trying to include more baselines for the camera-ready version: particularly, training the same RL policies using assets generated by URDFormer and Real2Code. We do expect these baselines to perform much worse given the poor quality of generated assets (Fig. 7). These results are not currently available because whereas our method can articulate real objects out of the box, other baselines require heavy curation (Appendix A.4) so it's difficult to curate and include them in time on top of all other experiments we've included in the global response to all reviewers. Once these baselines are finished, we will include a quantitative comparison table between these 4 baselines (ours, human-annotated, Real2Code, URDFormer) that the reviewer might expect.
>
> Please feel free to let us know if there's any more analysis or experiments that can be reasonably run before the deadline that you'd like to see to improve your rating or confidence.
>
> Thank you.

---

> > ### Author Response · Authors · 2024-12-02
> >
> > Dear Reviewer SpUi,
> >
> > Thank you for your positive feedback and helping us improve the paper!
> >
> > As a friendly reminder, today is the last day that reviewers can post a message to the authors. Please do not hesitate to let us know if there is any more question or issue that we can address.
> >
> > Thank you,
> >
> > Authors

---

### Author Response · Authors · 2024-11-22
**Summary of reviews and our revision**

We present Articulate-Anything, an "original and significant" (SpUi), "interesting" (yrhN), "intriguing and highly effective" (Bhiq) method for automatically articulating 3D objects. To our knowledge, this is the only method that can (1) articulate from many inputs, including text, images, and videos; (2) leverage high-level program synthesis; and (3) visual feedback for self-improvement, producing high-quality assets and achieving state-of-the-art performance, overtaking prior works by nearly sixfold.

We'd like to thank all reviewers for their suggestions and have incorporated their feedback in the paper.
- Robotic application [SpUi, yrhN]: Compared to human labeling, we show that Articulate-Anything saves, on average, 40.58 minutes of annotation time per task. The policies trained on our automatically articulated assets can be deployed to a real robotic system, successfully completing all tasks. This can enable future applications in more massively scalable robot training. Please see the updated Sec. 6 and [website](https://iclr-2025-4168.github.io/).
- Mesh generation [bBwZ, yrhN]: our articulation components are general-purpose and can be integrated with any mesh acquisition method. In addition to the retrieval mechanism, we have included results on articulating meshes generated by Rodin, a diffusion model. Please see updated Fig. 7, 22 and the [website](https://iclr-2025-4168.github.io/). We also quantify our superior performance using Chamfer distance in Table 2 (appendix A.5).
- Incontext-learning and ablation on the base VLM [VbEW]: We have included Sec. 5.3, showing (1) our method exhibits in-context learning, and (2) is robust to the choice of VLMs.
- Additional visualizations [SpUi, bBwZ]: We have included in the Appendix A.5 the visualization of the error distribution, and break-down by object categories.

---

### Meta-Review · Area_Chair_UJ7s · 2024-12-25

**Metareview:**

This work introduces a novel approach to automate the articulation of complex objects from multi-modal inputs (text, images, videos). Four reviewers have expressed positive feedback, while one reviewer has raised some concerns. The author has done a good job during rebuttal. The Area Chair finds the work interesting and recommends it for publication at NeurIPS 2024. The reviewers have highlighted some valuable concerns that should be addressed in the final camera-ready version. The authors are encouraged to incorporate the necessary revisions in the final submission.

**Additional Comments On Reviewer Discussion:**

NA

---

### Decision · Program_Chairs · 2025-01-22

Accept (Poster)